# 3D electron diffraction–the missing slice completing nanoscale analysis of organic solar cells in TEM

Irene Kraus [1], Mingjian Wu [1] ✉, Stefanie Rechberger[1], Johannes Will [1], Santanu Maiti[2], Konstantin Dengel [1], Andreas Kuhlmann [3,4,5], Marten Huck[3,4,5], Larry Lüer [6,7], Florian Bertram[8], Hans-Georg Steinrück[3,4,5], Tobias Unruh [2], Christoph J. Brabec [6,7] & Erdmann Spiecker [1] ✉

Optimizing the performance of organic solar cells hinges on a comprehensive understanding of their nanostructures, yet traditional characterization methods often fall short, delivering incomplete structural snapshots. We introduce elastically filtered 3D electron diffraction as technique to bridge full reciprocal- and real-space structural analysis within a single transmission electron microscope. Using model bulk heterojunction DRCN5T:PC$_{71}$BM, 3D electron diffraction reproduces key structural parameters obtained from grazing-incidence wide-angle X-ray scattering, including lattice spacings, coherence lengths, and mosaicity, while also providing true in-plane access and direct registration with high-resolution imaging, diffraction imaging and nano-spectroscopy on the same sample. Application to another archetypal blend, P3HT:PC$_{71}$BM, demonstrates the generality of the method. Our findings underscore the transformative potential of 3D electron diffraction, particularly in analyzing beam-sensitive organic thin films. The method enables correlative structural characterization of organic solar cells and opens pathways for application to a wide range of other nanostructured materials.

Organic solar cells (OSCs) have seen remarkable advancements, with power conversion efficiencies now exceeding 20%[1,2], largely driven by innovations in molecular design[2–6], material engineering and processing optimization[7–13]. In the most widely adopted OSCs, namely bulk-heterojunction (BHJ), donor and acceptor materials are intimately blended in the active layer at the nanoscale to facilitate efficient charge generation and transport. The performance of these devices is inherently linked to their structural organization and nanomorphology, as the molecular assembly and ordering dictate efficiencies of the primary processes in OCSs, often with contradicting requirements. For example, large domains facilitate charge collection but limit exciton dissociation. This trade-off can be alleviated by domains of high anisotropy, showing efficient exciton dissociation and charge collection simultaneously[14]. This emphasizes that detailed knowledge of nanoscopic molecular ordering and morphology is needed to understand optoelectronic performance. Moreover, the microstructural evolution of OSCs is sensitive to processing conditions, with parameters such as solvent choice, annealing, and deposition techniques critically

[1]Institute of Micro- and Nanostructure Research & Center for Nanoanalysis and Electron Microscopy (CENEM), Friedrich-Alexander-Universität Erlangen-Nürnberg, IZNF, Erlangen, Germany. [2]Institute for Crystallography and Structural Physics, Friedrich-Alexander-Universität Erlangen-Nürnberg, Erlangen, Germany. [3]Institute for a Sustainable Hydrogen Economy (IHE-1), Forschungszentrum Jülich GmbH, Jülich, Germany. [4]Institute of Physical Chemistry, RWTH Aachen University, Aachen, Germany. [5]Department of Chemistry, Paderborn University, Paderborn, Germany. [6]Institute Materials for Electronics and Energy Technology (iMEET), Friedrich-Alexander-Universität Erlangen-Nürnberg, Erlangen, Germany. [7]Helmholtz Institute Erlangen-Nürnberg for Renewable Energy (HIERN), Forschungszentrum Jülich GmbH, Erlangen, Germany. [8]Deutsches Elektronen-Synchrotron DESY, Hamburg, Germany. ✉e-mail: mingjian.wu@fau.de; erdmann.spiecker@fau.de

influencing the resulting morphology[8,15,16]. Given the strong structure-property relationship in OSCs, precise characterization of their molecular organization and nanoscale morphology is essential for understanding and optimizing device performance.

Grazing-incidence wide-angle X-ray scattering (GIWAXS) is an established and widely used technique for quantitatively characterizing the structure of OSCs, utilizing both laboratory and synchrotron X-ray sources. GIWAXS allows extraction of key structural parameters, including distances of inter-molecular packing, crystallite size, degree of ordering, and phase purity[17–19]. By leveraging the in-plane isotropy of BHJ films, GIWAXS simultaneously probes in-plane and out-of-plane molecular arrangements in a single measurement. This capability, along with in situ monitoring of structural evolution[20], makes GIWAXS a valuable tool for studying OSC microstructure. At the same time, as an ensemble-averaging method, GIWAXS cannot deliver real-space nanomorphology, local orientation heterogeneity, or direct chemical contrast, and it intrinsically misses the exact $q_z = 0$ plane (pure in-plane) due to the substrate horizon. These limitations motivate an integrated approach in which quantitative reciprocal-space sampling is registered with nanoscale imaging and spectroscopy, achievable only inside the modern transmission electron microscope (TEM) via 3D electron diffraction (3D ED) and associated diffraction-imaging modalities.

The scattering principles of X-rays and electrons are closely related. While X-rays interact with the atomic electron density[18], electrons scatter from the screened electrostatic potential of the nuclei. Using the Poisson equation, the atomic scattering factors for X-rays and electrons can be directly converted into each other, as described by the Mott formula[21]. The diffraction of both X-rays and electrons is governed by Bragg's law, thereby enabling, in principle, the extraction of equivalent structural information. TEM is well suited for the structural characterization of thin-film materials, including OSCs. Modern TEM integrates imaging, diffraction, and spectroscopy within a single instrument, enabling multi-modal analysis. For OSCs, TEM provides nanoscopic insights into phase separation, and donor-acceptor inter-mixing through elemental mapping with analytical signals obtained via inelastically scattered electrons and/or emitted characteristic X-rays[8,9,16,22–26]. It also enables seamless switching between reciprocal- and real-space analysis, correlating molecular ordering with nanomorphology. However, TEM investigations of OSCs face challenges, including beam sensitivity[27] and difficulties in obtaining out-of-plane structural information. While cross-sectional analysis via focused ion beam (FIB) preparation can address the latter[9], it introduces ion damage and limits the accessible sample area.

Among TEM-based diffraction techniques, 3D ED has emerged as a powerful tool for mapping three-dimensional reciprocal space with high accuracy and precision. Originally developed for the structural analysis of submicron-sized crystals that are too small for conventional single-crystal X-ray diffraction, 3D ED has revolutionized crystallography. Automated data acquisition approaches[28–32] and ab initio structure determination routines[33–36], have enabled high-throughput workflows, facilitating the study of beam-sensitive materials such as metal-organic frameworks and pharmaceuticals. Despite these successes, 3D ED remains underutilized for polycrystalline and partially ordered materials, particularly in OSC research. This limited adoption can be attributed to several challenges: (1) the strong inelastic scattering in OSCs necessitates elastic filtering, which is not always available, and (2) the high sensitivity of OSCs to electron irradiation requires careful optimization of experimental conditions to mitigate beam-induced damage. Yet these challenges do not constitute fundamental limitations to the applicability of 3D ED. With appropriate experimental optimization, 3D ED can provide detailed insights into the nanoscale structure of OSCs. We explicitly benchmark 3D ED against GIWAXS—the established structural reference in the field that has enabled seminal thin-film structure determinations[2,37,38]—to validate quantitative fidelity while revealing added correlative value.

3D ED fits naturally into the comprehensive, correlative characterization toolkit for organic semiconductor thin films in modern TEM, where a single instrument integrates imaging, analytical techniques, diffraction, diffraction imaging, and 3D ED. Here, we demonstrate the use of 3D ED as a quantitative technique for the structural characterization of organic thin films, adding 3D molecular ordering parameters, such as texture and mosaicity, to the extensive characterization of nanomorphology, in-plane molecule ordering, local molecule packing, and composition (cf. Fig. 1). This integration allows for a more complete characterization of structural and morphological features, which are crucial for optoelectronic processes in organic semiconductors[39]. Using a well-studied model BHJ system (DRCN5T:PC$_{71}$BM)[8,15,16,40,41], we develop a 3D ED workflow and validate its reliability by comparing results with GIWAXS from laboratory and synchrotron sources. By integrating 3D ED with TEM-based imaging, diffraction, and analytical capabilities, we position 3D ED as a complementary approach to GIWAXS for the comprehensive characterization of OSCs. Using the same model BHJ sample, we illustrate the range of multi-modal data obtainable from a single OSC specimen in the TEM (cf. acquired data in Fig. 1). Our results demonstrate how detailed molecular packing information from 3D ED, revealing molecular texture and mosaicity, can be directly correlated with these morphological features. Furthermore, applying 3D ED within a comprehensive TEM workflow reveals structural and morphological changes induced by thermal annealing in a second archetypal OSC system (P3HT:PC$_{71}$BM), demonstrating the broad applicability of our method. By contrasting the respective strengths and limitations of 3D ED and GIWAXS, we specifically underscore the complementary character of both methods. Furthermore, we discuss the potential of our 3D ED approach for future applications.

## Results

First, we compare the diffraction geometries and resulting reciprocal space sampling of 3D ED and GIWAXS. Using the well-studied model BHJ system (DRCN5T:PC$_{71}$BM), we then outline the 3D ED workflow and discuss the structural information available directly from analyzing diffraction patterns at multiple sample tilt angles. Next, we compare results obtained from 3D ED and GIWAXS on the same sample system, focusing on distinct intensity profiles extracted from the 2D patterns and the corresponding molecular ordering characteristics. We subsequently highlight the insights obtained by the correlative use of 3D ED with other TEM methods. Finally, we demonstrate the broader applicability of our method and its integration into a correlative workflow by resolving structural changes in the well-known P3HT:PC$_{71}$BM system, consistent with existing literature.

### Diffraction geometry and missing wedge

GIWAXS and 3D ED provide complementary approaches for probing the molecular ordering of thin films. For organic semiconductors ordering of the molecules in crystallites is typically described with respect to three directions, namely π-π stacking as [010], lamellar stacking as [100] and backbone direction as [001][19], which is also applied in the following (cf. inset in Fig. 2a).

Figure 2a–b schematically illustrate the characteristic diffraction geometries for GIWAXS and 3D ED, respectively. In GIWAXS, an X-ray beam with incident wavevector $\mathbf{k}_{i,GIWAXS}$ impinges on the sample at a grazing angle ($\alpha_{i,GIWAXS}$), enabling forward scattering at specific exit wavevectors ($\mathbf{k}_{f,GIWAXS}$). The diffracted intensities are recorded on a 2D detector and reflect in-plane ($q_x$, $q_y$) and out-of-plane ($q_z$) components of the scattering vector. An exemplary diffraction pattern of a BHJ sample is displayed in Fig. 2a. The choice of $\alpha_{i,GIWAXS}$ relative to the critical angles of the substrate and film optimizes surface sensitivity and reduces substrate contributions. Due to the finite incidence angle, in-plane information is only indirectly accessible unless high-energy X-rays are used in grazing incidence transmission mode[42] (the exemplary

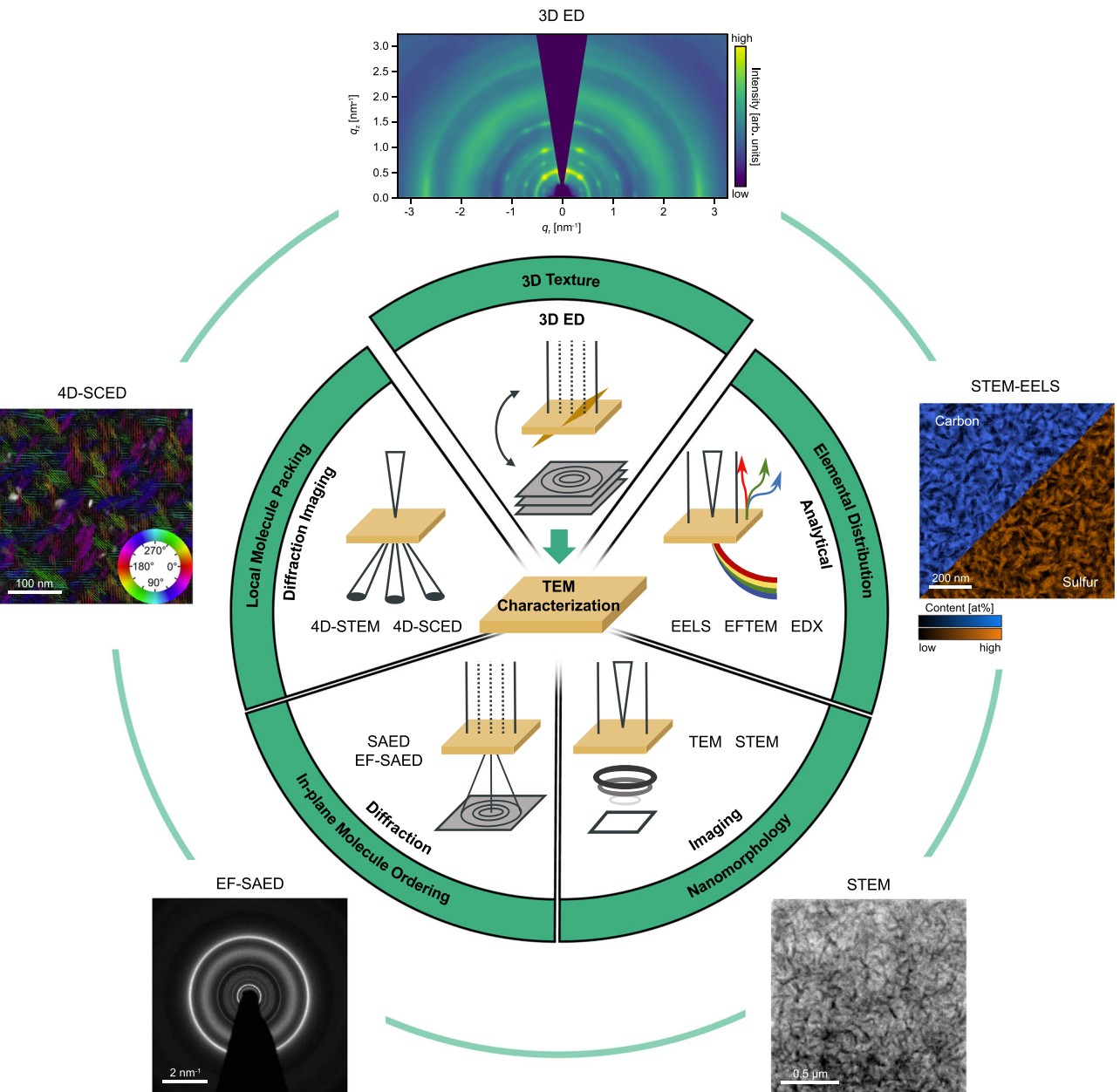

**Fig. 1 | 3D ED completing the structural characterization of organic semi-conductor thin films in TEM.** The central scheme illustrates the integration of 3D ED into the suite of analytical, imaging, diffraction and diffraction imaging methods offered by a single TEM instrument. Analytical modalities include electron energy loss spectroscopy (EELS), energy filtered TEM (EFTEM), and energy dispersive X-ray spectroscopy (EDX); imaging comprises conventional TEM methods or scanning TEM (STEM); diffraction includes selected-area electron diffraction (SAED), optionally energy filtered (EF-SAED); and diffraction imaging modalities capture 2D real and 2D reciprocal information, such as 4D-STEM or scanning confocal electron diffraction (4D-SCED). The addition of 3D ED provides insights into 3D texture and mosaicity, complementing already available information on chemical composition, nanomorphology, in-plane molecule ordering, and local molecule packing, thereby significantly enhancing the characterization capabilities of OSC thin films. Surrounding the scheme, the outer ring of images presents data measured on a single sample of the DRCN5T:PC$_{71}$BM OSC model system in the TEM, aligned with the corresponding TEM methods. This comprehensive characterization facilitates the direct correlation of real and reciprocal space information, linking 3D molecular packing with nanoscale morphology and composition.

crystallite in Fig. 2a is thus depicted inclined slightly, resulting in π-π stacking diffraction signal with out-of-plane component).

In electron diffraction, the incident beam with wavevector $\mathbf{k}_{i,TEM}$ propagates perpendicular to the film in a plan-view setup. 3D ED extends this approach by tilting the sample (and/or the incident beam[36]) across a range of angles ($\alpha_{TEM}$) (Fig. 2b) to sequentially capture slices of three-dimensional reciprocal space information as detailed in the following paragraph. The relationship between incidence angles in GIWAXS and 3D ED follows $\alpha_{TEM} = 90° - \alpha_{i,GIWAXS}$. The ring diffraction pattern shown exemplarily arises from diffraction contributions of many crystallites randomly oriented perpendicular

to $\mathbf{k}_{i,TEM}$ (the red $\mathbf{k}_{f,TEM}$ vector represents π-π stacking diffraction signals in Fig. 2b).

Reciprocal space sampling of 3D ED and GIWAXS is illustrated in Fig. 2c–f, using the Ewald sphere construction. The short electron wavelength (typically few pm) results in a large Ewald sphere, making reciprocal space sections nearly flat in 3D ED (Fig. 2c). Tilting the sample thus samples different slices of reciprocal space and progressively builds a three-dimensional dataset (Fig. 2d). At zero tilt, in-plane sample information is encoded. In GIWAXS, with its longer X-ray wavelength (on the order of Angstroms), a smaller Ewald sphere with noticeable curvature samples reciprocal space (Fig. 2c,e). Building up

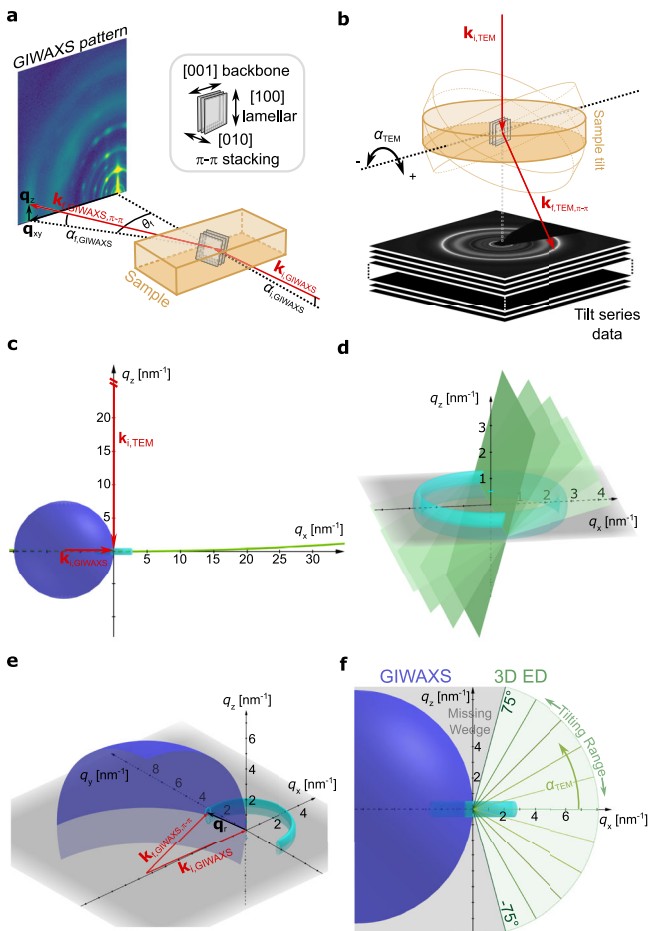

**Fig. 2 | Schematic of scattering geometries and reciprocal space sampling of 3D ED and GIWAXS. a** In GIWAXS, the X-rays with incident wavevector $\mathbf{k}_{i,GIWAXS}$ impinge the sample at a grazing angle $\alpha_{i,GIWAXS}$ and the diffracted signal is detected by a detector perpendicular to the sample plane (exit wavevector $\mathbf{k}_{f,GIWAXS}$). The $\mathbf{q}_{xy}$ reciprocal vector represents the in-plane diffraction angle $\theta_f$, the $\mathbf{q}_z$ vector represents out-of-plane angle $\alpha_{f,GIWAXS}$. The inset illustrates characteristic lattice distances of molecular packing in organic semiconductor crystallites: lamellar packing [100] and backbone direction [001] (~2-3 nm), and π-π stacking ([010], ~0.3-0.4 nm), typically perpendicular to [100]. **b** In 3D ED the sample is tilted by an angle $\alpha_{TEM}$ around a specified tilt axis determined by the sample holder and microscope stage. This varies the incidence angles of the electron beam on the sample (incident wavevector $\mathbf{k}_{i,TEM}$) and thereby enables acquiring tilt series datasets with a detector below the sample (exit wavevector $\mathbf{k}_{f,TEM}$). The typical diffraction angles in b are «1° due to the high electron energy. This illustration is exaggerated for better visualization. **c** The sizes of Ewald sphere in GIWAXS and 3D ED using typical experimental conditions differ. Corresponding incidence wavevectors $\mathbf{k}_i$ are marked. The turquoise ring schematically depicts the reciprocal space lattice of a textured OSC film with dominating edge-on π-π stacking (cf. Supplementary Fig. 1c). **d** Approximating the Ewald sphere for fast electrons as flat plane and acquiring a tilt series progressively samples the 3D volume of reciprocal space. The green planes represent the almost flat Ewald sphere cuts through reciprocal space. **e** 3D view of the Ewald sphere and Laue condition for GIWAXS case. $\mathbf{q}_r$ represents in-plane coordinates $q_x$ and $q_y$. **f** The comparison of 3D ED and GIWAXS reciprocal space sampling reveals the missing wedges of both methods using typical experimental conditions, quantifying the almost equal (in-)completeness of their reciprocal space sampling.

the 3D reciprocal space would require successive sampling, that is, repeated measurements, by rotating the sample about the $z$-axis. However, for BHJ samples, in-plane structural isotropy can typically be assumed to a good approximation. Consequently, a single measurement is sufficient, and rotation about the surface normal does not provide new reciprocal space information. Both methods exhibit missing regions in reciprocal space around the $q_z$-axis, that is, a

missing wedge, albeit for different reasons. In 3D ED, this arises from the restricted sample tilt range, whereas in GIWAXS it results from the finite size of the Ewald sphere and the limited range of incidence angles (cf. Fig. 2d,e). Leveraging in-plane isotropy of samples in GIWAXS, single $q_{xy-z}$ measurements can be converted to a $q_{rz}$ map of the reciprocal space[18,43]. This conversion reveals that the $q_z$-axis is not sampled as it seems in the originally detected pattern (cf. Fig. 2a), but that there is a curved missing wedge centered around and widening along the $q_z$-axis (see Supplementary Fig. 1a,b). This curved missing wedge can be reduced with higher energy X-rays, e.g., using synchrotron source. Using typical experimental conditions, Fig. 2f shows a comparison of the missing wedge from 3D ED and GIWAXS for the study of OSCs. The almost equal (in-)completeness of reciprocal space sampling of 3D ED and GIWAXS can be clearly seen. To facilitate a direct comparison of 3D ED and GIWAXS, we adopt the notation commonly used in ED for both techniques, where reciprocal lattice vectors are expressed as $|\mathbf{g}| = 1/d$ and wavevectors as $|\mathbf{k}| = 1/\lambda$.

## Electron diffraction tilt series and 3D ED

As previously shown in our studies[8,15,16,40,41], films of the model BHJ system DRCN5T:PC$_{71}$BM exhibit a 2D oblique crystal system after solvent vapor annealing, where the small molecule crystals exist in dominant texture of face-on and edge-on orientations[41]. For clarity, we refer to the π–π stacking direction as the normal to the π-conjugated plane of DRCN5T, which corresponds to the [010] direction and is perpendicular to both the lamellar stacking direction [100] and backbone direction [001][41].

Figure 3a illustrates the structure of the DRCN5T molecular crystal and structural information represented in 3D ED data, including different crystallite orientations, rotations and inclinations. Diffraction patterns were recorded over a tilt range of −78° to 80° with 1° steps (cf. tilt series in Supplementary Movie 1). To mitigate beam damage, each pattern was acquired on fresh area unexposed to electrons via systematic shifting of the sample (see Methods). Inelastically scattered electrons were filtered to enhance the signal-to-background ratio (SBR), as discussed in more detail later. Representative ED patterns at tilt angles of 0°, 30°, and 75° (cf. Fig. 3b,c) already reveal the underlying texture of the reciprocal lattice. At 0°, the in-plane reciprocal space ($q_{xy}$ plane) is sampled. The ED pattern reveals isotropic diffraction rings characteristic of polycrystalline thin films with random in-plane orientations[39]. At higher tilts, azimuthal intensity modulations appear, indicating preferred crystallite orientations. The diffraction features arise from three fractions of molecular ordering: edge-on, face-on, and isotropic orientations. At 0°, the outer diffraction ring ($q \approx 2.73$ nm$^{-1}$) arises from π–π stacking ((010) lattice planes) of edge-on crystallites with a random in-plane orientation (cf. Fig. 3 rotation angle $\Phi_e$), while the low-$q$ ring near $q \approx 0.53$ nm$^{-1}$ originates from lamellar stacking ([100] direction) in predominantly face-on domains (cf. Fig. 3 rotation angle $\Phi_f$). Upon tilting, an intense segment appears along the $q_y$-axis of the π–π stacking ring—attributable to edge-on crystallites—while weaker intensity from isotropic crystallites persists at other azimuth angles (cf. Fig. 3 angle $\chi$). The edge-on crystallites would exhibit an intense segment of the lamellar packing ($|\mathbf{q}| \approx 0.53$ nm$^{-1}$) along $q_z$ at 90° (cf. Supplementary Fig. 1c), but due to their low mosaicity (i.e., high degree of alignment) and the missing wedge, this feature is not sampled within the tilt series. These results highlight that tilt-series diffraction patterns provide access to the 3D molecular ordering in OSC thin films, as demonstrated for other systems as well[25]. For further quantitative evaluation, the tilt-series was reconstructed into a 3D reciprocal space volume (cf. ortho-slice view in Fig. 3d and further representation in Supplementary Fig. 2 and Supplementary Movies 2 and 3), including normalization for tilt-dependent intensity variations (see Methods). The reconstructed volume clearly reveals the superimposed fiber-textured lattices corresponding to face-on and edge-on domains. For direct comparison with GIWAXS, the volume is

then integrated azimuthally around the $q_z$-axis, yielding a $q_{rz}$ map. This step enhances the signal-to-noise ratio (SNR) but averages over multiple, assumingly in-plane homogeneous sample positions due to the data acquisition scheme. Despite this, the total probed area in 3D ED (~5000 μm² range) remains significantly smaller than in GIWAXS (on the order of few mm² range).

## Comparison of 3D ED and GIWAXS results

Figure 4a compares $q_{rz}$ maps from 3D ED and GIWAXS, revealing matching diffraction rings and reflections. The good agreement confirms that 3D ED yields structural information comparable to GIWAXS. Based on prior indexing of GIWAXS data[41], reflections were assigned to DRCN5T crystallites. The amorphous halo at $q_r \approx 2$ nm⁻¹ originates from PC$_{71}$BM, while sharp rings with azimuthal segments stem from DRCN5T crystallites. A fully indexed $q_{rz}$ map is shown in Supplementary Fig. 3. The different shapes of the missing wedge in each dataset reflect respective limitations of the methods: limited tilt range in 3D ED and Ewald sphere curvature in GIWAXS.

To quantify the agreement, we compared in-plane diffraction profiles from the 0° ED pattern and the corresponding GIWAXS linecut (Fig. 4b). The profiles align well, with 3D ED exhibiting a higher SNR despite probing a much smaller volume under low-dose conditions. The 0° ED pattern samples the $q_{xy}$ plane directly, whereas GIWAXS would require in-plane rotation to probe in-plane anisotropies, if any. Additionally, in GIWAXS, the sample horizon casts a shadow on the detector near $q_z = 0$[18], necessitating linecuts at slightly displaced positions ($q_z = 0.01$–$0.06$ nm⁻¹). This causes peak shifts and broadening in GIWAXS, which lead to underestimation of the crystal coherence length (CCL), due to geometric effects, though minimal in this study (Supplementary Fig. 4, Supplementary Table 1).

We extracted peak positions and FWHMs of the (100) lamellar stacking and (010) π–π stacking peaks. CCLs were calculated using the Scherrer equation (shape factor = 1). As shown in Table 1, 3D ED and GIWAXS yield consistent quantities of CCLs: ~5 nm for π–π stacking and ~20 nm for lamellar stacking, consistent with prior reports on the leaf-like morphology of this system[8]. Nevertheless, it is important to interpret CCLs cautiously in terms of crystallite sizes, as they can be affected by other factors, including strain, inhomogeneities, and instrumental broadening[18,39].

We further evaluate rotationally averaged $q_{rz}$ profiles (Fig. 4c). While this is straightforward in 3D ED, surface reflection and refraction in GIWAXS can lead to virtual peak shifts for out-of-plane reflections[18,44]. To account for these effects, the GIWAXS $q_r$-axis was rescaled by 1.5% (Supplementary Fig. 5), yielding close agreement between the two methods.

Mosaicity, which quantifies crystallite alignment, was extracted for different reflections from constant-$q$ profiles ($\chi$-cuts) in the $q_{rz}$ maps for further quantitative comparison. As summarized in Table 1, both methods yield consistent mosaicity of ~6° for most reflections and the angular positions agree well. However, the (10-1) peak from 3D ED shows a slightly larger $\chi$ width (~8°). Following geometric considerations, the mosaicity of (002) and (10-1) peaks is related and should be equal (cf. Supplementary Fig. 6a). This deviation may be due to sample inhomogeneity or film buckling during grid preparation. Since diffraction data were collected at different sample positions (to avoid beam damage), local variations can contribute to the apparent mosaicity. Furthermore, the 1° tilt step limits angular resolution in 3D ED and could be improved with finer steps. We note while GIWAXS measures only one (curved) slice of 3D reciprocal space, the 3D ED data presented in Fig. 4a is integrated from full 3D reciprocal space. We also analyzed the mosaicity of the in-plane π–π stacking peak with respect to out-of-plane inclination (Supplementary Fig. 6b). Both methods again yield consistent results (~22.5°), reinforcing the complementary nature of 3D ED and GIWAXS.

In addition to laboratory-source GIWAXS, we performed synchrotron-based GIWAXS at P08 (DESY, Hamburg). The results again are in good agreement with 3D ED (Supplementary Fig. 7). Compared to the lab source (16 h) and 3D ED (~3 h currently via manual acquisition), synchrotron measurements were much faster (~10 s), enabling in situ studies[20,45]. Moreover, the higher and tunable photon energies reduce the GIWAXS missing wedge (cf. Supplementary Fig. 7c). However, in-plane signals still exhibit lower SNR in GIWAXS than in 3D ED, as only a small subset of crystallites contribute to in-plane Bragg peaks in GIWAXS. This is well-known in GIWAXS analysis and accounted for using a sin($\chi$) correction during analysis[18,46].

## Correlative TEM analysis

A major strength of 3D ED is its integration in modern TEM platforms, which facilitates seamless switching between diffraction, imaging, and spectroscopy methods for comprehensive and correlative characterization (cf. Fig. 1). Notably, this correlative real-space/diffraction approach allows us to link morphological features with molecular packing orientation. Figure 5 shows a correlative dataset acquired from a single sample in TEM, comprising STEM-EELS, 4D-SCED and 3D ED. STEM-EELS provides elemental mapping and reveals donor-acceptor phase separation through their chemical composition: the small molecule contains more sulfur, while the fullerene acceptor is richer in carbon. The corresponding S-L and C-K maps obtained via model-based fitting[47], resolve leaf-shaped donor domains (cf. Fig. 5a, see Supplementary Note for details on evaluation). Additionally, 4D-SCED analysis reveals the presence and orientations of edge-on and face-on crystallites relative to the morphology of domains through the in-plane π-π stacking and (100) peaks (cf. Fig. 5b). The π-π stacking orientation map shows that the leaf-shaped regions consist of edge-on oriented DRCN5T crystallites with their π-π stacking plane normal aligned along the domain's short axis, as schematically depicted in the inset. In contrast, smaller or more rounded domains mainly host more face-on oriented crystallites[40]. This specific correlation between domain shape and molecular packing (edge-on vs. face-on) is significant; it indicates that solvent-vapor annealing induces alignment of the crystalline domains with the mesoscopic phase separation structure. Such alignment can influence charge transport pathways: edge-on oriented domains (π-π stacking direction in-plane) may facilitate in-plane transport across larger distances, whereas face-on domains promote vertical transport[48]. Furthermore, the PC$_{71}$BM also contributes to the 4D-SCED diffraction signal featuring an amorphous halo, as already discussed for the SAED patterns. The corresponding signal is mapped by applying an annular virtual detector, revealing the acceptor distribution via underlying diffraction data, as an alternative way to reveal the acceptor distribution via the analytical information in STEM-EELS.

By combining 3D ED (cf. Fig. 5c) with complementary analytical and (diffraction-)imaging methods, we directly reveal how nanomorphology and molecular orientation coincide in the BHJ film, an insight that pure GIWAXS, lacking real space information, cannot provide. In our case, we find that each DRCN5T leaf-like domain (~80 nm long) contains multiple smaller crystallites (~20 nm, per Scherrer analysis, cf. Table 1), indicating a mosaic structure within domains (cf. Fig. 5b inset). This suggests the presence of grain boundaries within domains, which relate to trap states and charge carrier recombination[8]. Moreover, 3D ED is more radiation dose-efficient and provides access to the degree of crystallite alignment, particularly out-of-plane, capturing inclined crystallites that deviate from strict edge-on or face-on orientation. The dataset reveals an isotropic fraction of crystallites, and deviations from strict orientation described as mosaicity (cf. Table 1 and Supplementary Fig. 6). Pinpointing these relationships (domains vs. crystallites, orientation vs. shape) and their close connection to optoelectronic processes underscores the value of a correlative TEM approach for understanding structure-property relationships in OSCs.

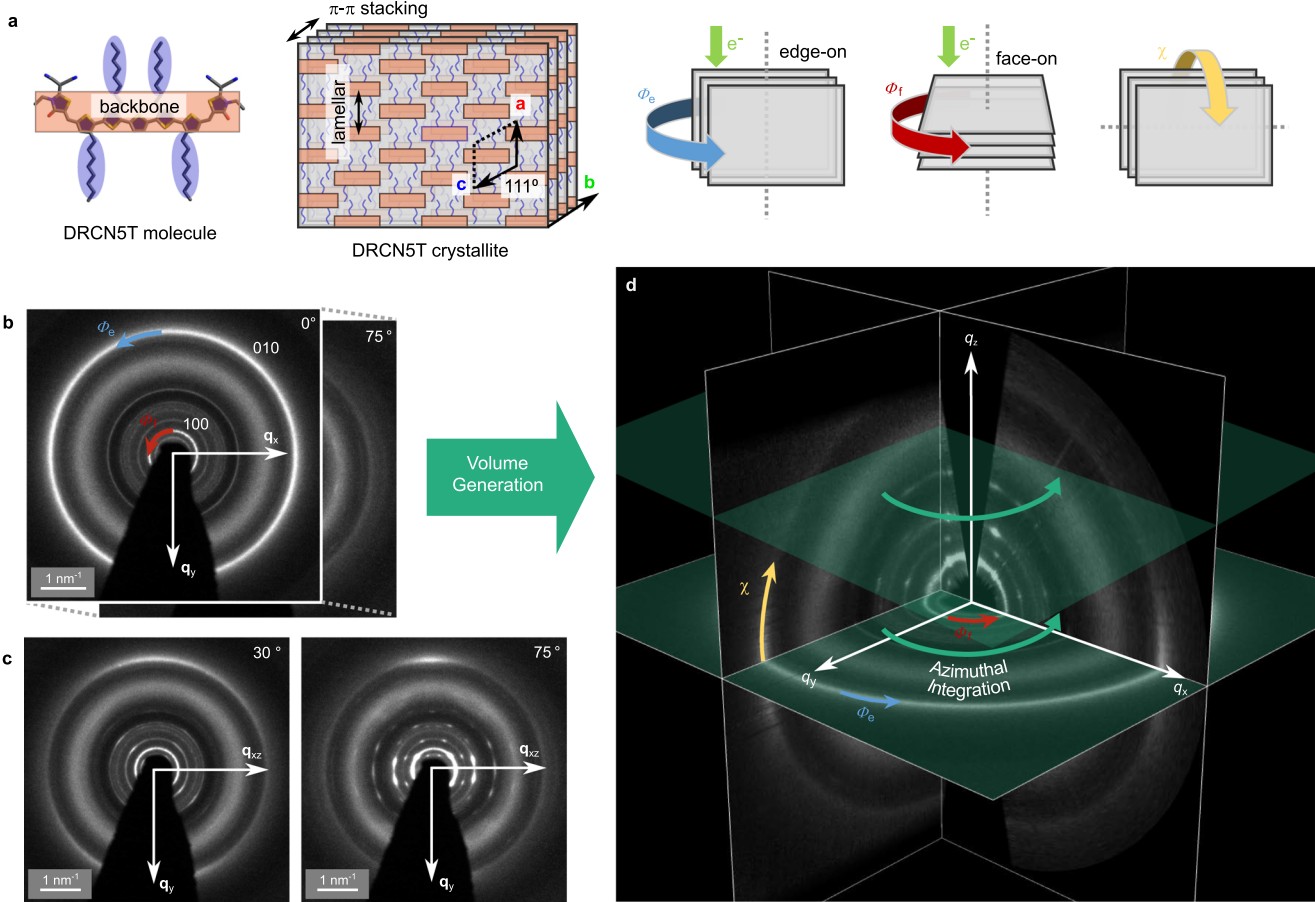

**Fig. 3 | Structural character of model molecular crystal and corresponding reciprocal space features extracted from 3D ED. a** The molecular arrangement of DRCN5T is exemplary shown for an edge-on crystallite. The molecule structure was simplified to maintain clarity (backbone orange, side-chains blue) and the oblique unit cell coordinates (**a**, **c** and **b**) are marked, where **a** represents lamellar stacking, and **b** π-π stacking. Different crystallite orientations and inclinations with respect to the electron beam are represented in the 3D ED data as marked accordingly: in-plane rotation of edge-on ($\Phi_e$, blue arrow) and face-on crystallites ($\Phi_f$, red arrow), and inclination ($\chi$, yellow arrow). **b** A tilt series from −78° to 80° was acquired in 1° steps. The pattern recorded at 0° sample tilt angle (in-plane, $q_{xy}$ plane) indicates the presence of both edge-on and face-on crystallites with random in-plane orientation

($\Phi_e$ and $\Phi_f$). **c** The representative patterns from the tilt series at 30° and 75° sample tilt (representing $q_{xyz}$ coordinates) reveal the texture of the system, manifested by intense diffraction ring segments. The bright segments of the outermost π−π stacking (010) ring appearing around the $q_y$-axis represent edge-on crystallites, whereas the otherwise weaker, continuous (010) ring indicates a fraction of isotropic crystallites (random inclination, $\chi$). **d** Out of the tilt series the three-dimensional reciprocal space volume is reconstructed, incorporating diffraction information of the differently oriented crystallites. The ortho-slice $q_y$-$q_z$ plane is offset from $q_x = 0$ to avoid the missing wedge. For comparison to GIWAXS an azimuthal integration is performed (see Fig. 4a).

## 3D ED resolving structural evolution

To demonstrate the general applicability of 3D ED and its integration in a correlative TEM workflow, we further apply it to study the archetypal P3HT:PC$_{71}$BM blend and unravel the structural details before and after thermal annealing (TA). 3D ED analysis of P3HT:PC$_{71}$BM films after TA reveals an increase in P3HT (100) lamellar stacking order compared to before TA, evidenced by the more clearly defined (100) ring and the appearance of higher order rings (cf. 3D ED $q_{rz}$ maps in Fig. 6a,b and 0° rotational average and spherical average in Supplementary Fig. 8a–c). Moreover, even the untreated sample shows texture, consisting of edge-on, face-on and isotropically oriented crystallites, as indicated by the azimuthal intensity of the (100) and π-π stacking rings (see azimuthal $\chi$-cuts in Supplementary Fig. 8d,e). The thermally annealed sample exhibits a comparable texture but shows sharper and more intense peaks, indicative of larger crystallite sizes, and a higher degree of crystallinity. 3D ED data allow not only CCL analysis of in-plane structure characters based on the 0° diffraction pattern, but also CCLs near out-of-plane direction by analyzing intensity profiles near the missing wedge (cf. intensity profile extracted at 74° relative to $q_r$ axis in Supplementary Fig. 8a,f). Such a CCL analysis indicates a clear

crystallite growth upon TA, preferably along [100] direction, while very little along π-π stacking direction, for both in-plane and out-of-plane direction (cf. Supplementary Table 2). This depicts a preferential growth along [100] for both face-on and edge-on crystallites, the former one extends in-plane, the latter one out-of-plane. We further elaborate this with 4D-SCED experiments and analysis on the same samples to reveal the in-plane structural changes and correlate to true real-space morphology. In the untreated sample, 4D-SCED analysis using the π-π stacking (010) signal reveals many small edge-on crystalline domains with size of only few probing pixels (cf. Fig. 6c). For face-on domains only very few domains over an area of few hundred thousand nm$^2$ could be reliably identified, which are just 2-4 probing pixels in size. We note that dense real space sampling ( ~ 2.7 nm pixel$^{-1}$) and thus higher dose ( ~ 5 e$^-$ Å$^{-2}$) was applied, which implies certain beam damage, although around the evaluated threshold, and very low diffraction signal to background was observed due to the tiny crystallites' contribution within the probed sample volume. Furthermore, we noticed that the (100) ordering is more beam sensitive than the π-π stacking, (010) signals in P3HT (cf. Supplementary Fig. 9, and Supplementary Table 3). In the annealed sample, lots of larger elongated

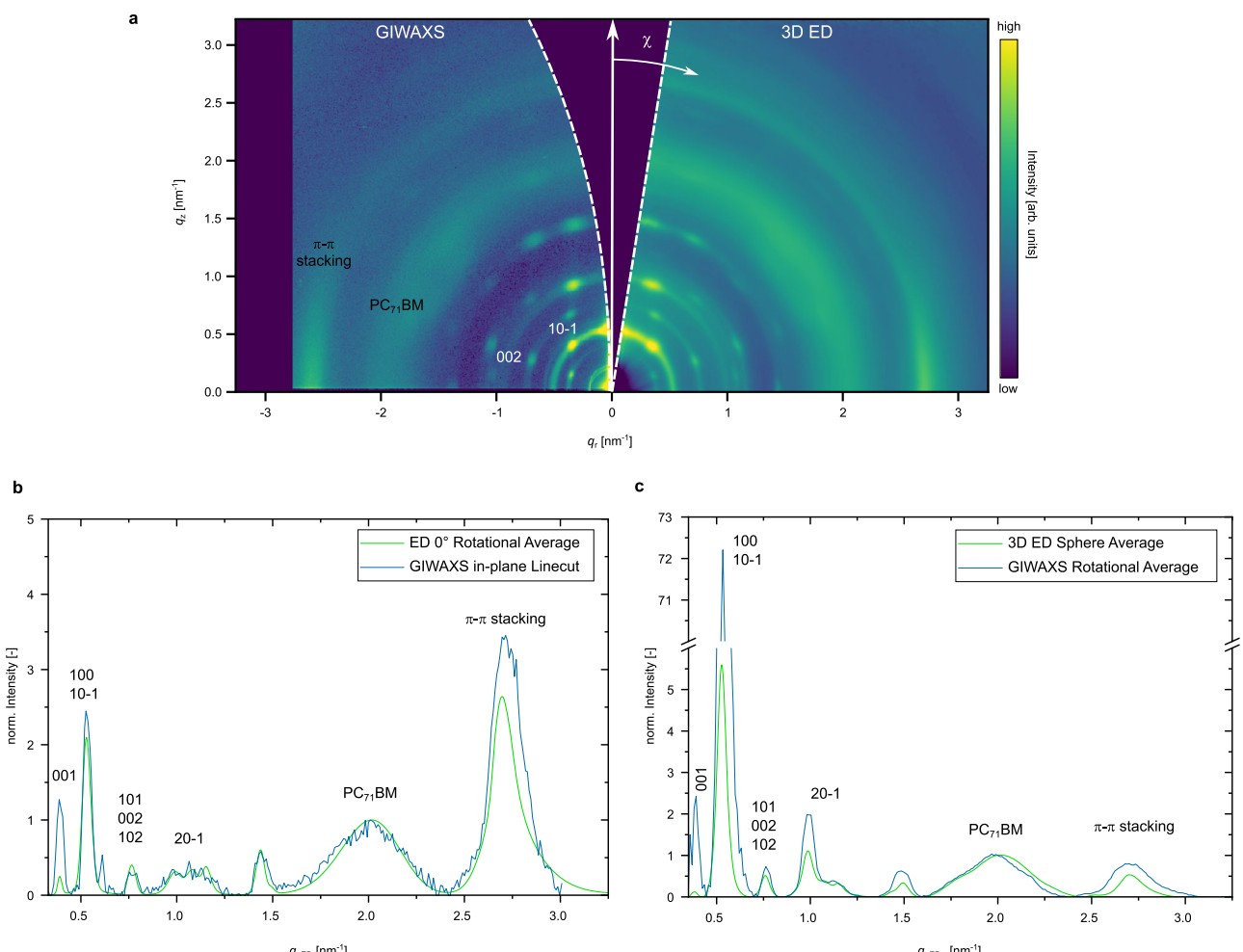

**Fig. 4 | Comparison of 3D ED and GIWAXS data. a** Qualitative overlay of 3D ED (right) and GIWAXS (left) $q_{rz}$ reciprocal space maps with exemplary indexed peaks. The azimuthal direction corresponds commonly to the coordinate $\chi$. Missing wedges along $q_z$-axis of both methods are marked by white dashed lines. A more quantitative comparison is based on different profiles extracted from the maps. **b** A GIWAXS in-plane linecut corresponds to the 0° ED pattern rotationally averaged. In **c** rotational averages of the $q_{rz}$ maps from both methods are compared. All shown profiles are background subtracted and normalized with respect to the PC$_{71}$BM peak.

domains (in projection) are apparent in both face-on and edge-on orientation. The edge-on domains show an elongation along the π-π stacking direction (correspondence of image and diffraction shown as inset in Fig. 6d and scheme in Fig. 6e). More face-on domains could be revealed after TA (cf. Fig. 6d, Supplementary Fig. 9 and Supplementary Table 3) than before, extending along the lamellar [100] direction, in agreement to CCL analysis from 3D ED. For both face-on and edge-on elongated domains, no ordering along the short fiber axis could be observed, which could be related to structural disorder along [001] backbone direction[49] or rotation of crystallites (towards out-of-plane). Furthermore, STEM and EFTEM imaging reveal the coarsening of nanomorphology and enhanced phase separation upon thermal annealing (cf. Supplementary Fig. 10). Those findings are consistent with the enhanced crystallinity and phase segregation known to improve device performance in this system[50–52]. The fact that 3D ED and its integration in a correlative TEM workflow can quantitatively capture these structural and morphological changes confirms that our approach is general and can be used to study processing-structure-property relationships in organic photovoltaic materials.

## Discussion

While both 3D ED and GIWAXS provide consistent information on the 3D molecular ordering of our DRCN5T model system, each method has

distinct strengths and limitations. This motivates a complementary use for comprehensive structural characterization. In the discussion below, we first address their common challenges before focusing on their specific advantages. Drawing on the demonstration of our approach on a second OSC system, we then outline practical considerations to facilitate broader use and end with a brief summary.

A central challenge for both 3D ED and GIWAXS is the radiation sensitivity of molecular semiconductors, which restricts the dose rate and total dose budget, thereby limiting the achievable SNR for a given sample volume[40]. This limitation is commonly regarded as more pronounced in ED, owing to the stronger interaction of electrons with matter and the much smaller illuminated area (μm² vs. mm²). The cross-section for inelastic scattering in organic thin films, the primary source of radiation damage, is several orders of magnitude higher for fast electrons than that for the hard X-rays used in GIWAXS. However, it is important to note that not only the inelastic scattering, but also the elastic scattering of electrons is significantly stronger than that of X-rays. This enhanced elastic interaction is the key factor that enables the detection of diffraction signals from such small sample volumes. Additionally, inelastic scattering signals play a significant role in ED by contributing to the background at small scattering angles. While these signals are generally neglected in GIWAXS, elastic filtering is essential in ED to suppress the pronounced inelastic scattering background at

**Table 1 | Comparison of structural ordering parameters determined from 3D ED and GIWAXS**

| Method | 3D ED | GIWAXS |
|---|---|---|
| In-plane diffraction | | |
| (100) position [nm$^{-1}$] | 0.5307 ± 0.0004 | 0.5325 ± 0.0013 |
| (100) FWHM [nm$^{-1}$] | 0.0532 ± 0.0010 | 0.0470 ± 0.0030 |
| π-π position [nm$^{-1}$] | 2.7315 ± 0.0029 | 2.7202 ± 0.0038 |
| π-π FWHM [nm$^{-1}$] | 0.1964 ± 0.0051 | 0.1922 ± 0.0099 |
| CCL (100) [nm] | 18.80 ± 0.35 | 21.26 ± 1.34 |
| CCL π-π [nm] | 5.09 ± 0.13 | 5.20 ± 0.27 |
| χ-cuts | | |
| (10-1) FWHM [°] | 8.12 ± 0.06 | 6.33 ± 0.33 |
| (10-1) χ Position [°] | 40.86 ± 0.03 | 40.81 ± 0.14 |
| (002) FWHM [°] | 6.84 ± 0.11 | 6.35 ± 0.36 |
| (002) χ Position [°] | 69.32 ± 0.04 | 68.77 ± 0.12 |
| In-plane π-π stacking FWHM [°] | 22.18 ± 0.18 | 23.14 ± 0.77 |

The crystal coherence lengths for the (100) and π-π stacking peaks were determined based on in-plane intensity profiles. Angular positions and azimuthal widths of different peaks were extracted from azimuthal χ-cuts of the $q_{rz}$ maps.

small scattering angles, thus improving the SBR in scattering angles relevant to organic crystals with large unit cells. In this work, we demonstrated that inelastic scattering background and beam damage can be effectively mitigated in 3D ED by combining energy filtering with a targeted workflow (fresh regions for each pattern), resulting in significantly improved SBR and SNR. We also examined the dose limits for our materials: notably, for DRCN5T the π-π stacking (010) reflection vanishes after a cumulative electron dose of ~ 5 e$^-$ Å$^{-2}$, whereas the lamellar (100) reflection persists up to doses exceeding 30 e$^-$ Å$^{-2}$ [40]. For P3HT, both π-π and lamellar stacking reflections, vanish at a critical dose between ~ 5 and 15 e$^-$ Å$^{-2}$ (cf. Supplementary Fig. 9 and Supplementary Table 3). We therefore designed our ED data collection to remain well below the ~ 5 e$^-$ Å$^{-2}$ threshold per area (see Methods), thereby ensuring the fidelity of the structural information. Accessing out-of-plane information near the $q_z$-axis remains challenging for both methods. Although increasing X-ray energy (GIWAXS) or tilt range (3D ED) improves access toward $q_r = 0$, obtaining pure $q_z$ information typically requires additional measurements, such as varying the incidence angle in GIWAXS or preparing cross-sectional TEM samples. On the other hand, pure in-plane ($q_{xy}$ plane) information is readily accessible in ED due to the transmission geometry (at 0° tilt angle), but remains fundamentally inaccessible in GIWAXS owing to the sample horizon and finite incidence angle. However, these limitations are largely mitigated in OSC layers, which typically exhibit broad mosaicity, on the order of several degrees[53], as demonstrated for the DRCN5T:PC$_{71}$BM system. Further experimental challenges include instrumental broadening (e.g., due to finite beam footprint in GIWAXS on the sample surface due to grazing incidence[18,44,54]) and reflection/refraction effects that cause peak shifting and splitting along $q_z$. While these effects complicate analysis, they can also provide additional information, such as refractive index and electron density. Nonetheless, they must be carefully considered when extracting lattice parameters from GIWAXS data[44].

Despite such limitations, both techniques offer comprehensive and complementary capabilities for studying OSC thin films from spatially averaged regions of interest (μm² to mm²). GIWAXS excels in its straightforward sample preparation, allowing direct measurements on functional thin films deposited on standard substrates (e.g., glass or silicon). Short acquisition times, especially at synchrotron sources, enable in situ studies during film formation or post-treatment (e.g., solvent vapor or thermal annealing). High-energy X-rays mitigate beam damage and reduce the missing wedge, while correlative methods like

X-ray reflectometry or photoluminescence provide additional insights into structural and electronic properties[20]. Moreover, depth-profiling through incidence angle variation enables investigation of vertical inhomogeneities and stratification effects, which are critical for OSC performance[18]. An emerging direction in GIWAXS involves the quantitative analysis of Bragg peak intensities to determine atomic positions within the unit cell. This requires consideration of multiple correction factors—Lorentz effects, absorption, transmission, solid angle, and X-ray polarization[43]. While similar analysis is established in 3D ED for single crystals[36,55,56], a robust framework for polycrystalline materials such as OSC thin films still needs to be explored. Conversely, 3D ED offers several advantages. Beyond its primary strength—compatibility with a wide range of correlative TEM-based characterization techniques (Figs. 1, 5) and the corresponding insights into critical structural and morphological features—it delivers a high SNR, particularly along in-plane directions, providing access to structural information that GIWAXS cannot directly access. Moreover, the technique allows rapid tuning of the $q$-range through flexible camera length adjustments and is capable of resolving spatial inhomogeneities and in-plane texture on the micrometer scale, owing to its adjustable field of view. In this context, we note that, prospectively, synchrotron X-ray scattering offers a complementary route to reveal such inhomogeneities by employing a (sub-)micrometer X-ray beam in (perpendicular) transmission geometry—analogous to plan-view ED in TEM. Single-shot transmission X-ray diffraction patterns from such small areas have recently been demonstrated for a donor–acceptor polymer film, utilizing a free-electron laser nano-beam with a full width at half-maximum of ~ 250 nm[57]. However, a direct correlation of this local yet still averaging information with spatial maps of crystallite orientation or molecular order at the nanometer scale can only be achieved in the TEM, employing low-dose techniques, such as 4D-STEM[58], 4D-SCED[40,59] or HRTEM under cryogenic temperatures[57].

We now turn to practical considerations for broader use of the 3D ED approach, beginning with in situ implementations. In situ studies involving temperature and gaseous environment are increasingly being adopted, facilitated by chip-based devices and advanced holder systems that offer a high tilting range suitable for tomography[60,61]. These setups, although designed for real-space observations, can be used for in situ 3D ED studies as well. Depending on the required temporal resolution, high-quality datasets can be acquired at multiple temperatures to probe structure–temperature relationships. For higher temporal resolution, the number of tilt angles can be reduced. Our recent investigation of PM6:Y-series OSC thin films demonstrated that even a two-angle acquisition (e.g., 0° and 75°) reveals valuable structural information beyond conventional plan-view imaging[25].

The possibility of capturing sparse snapshots of 3D reciprocal space is not only a valuable approach for achieving higher temporal resolution in in situ studies, but also for increasing spatial resolution of our 3D ED approach: in our demonstration, the 3D ED data represent a spatially averaged structure (covering ~ 5000 μm²), as the tilt series were distributed across the sample to minimize beam damage at any single location. Since homogeneity was experimentally confirmed in our OSC sample systems, this averaging approach is justified. For application in OPV homogeneity of the active layer is generally required and presumed. Importantly, 3D ED is in principle sensitive to local inhomogeneities and could therefore serve as a valuable tool to identify and evaluate them. However, achieving spatial mapping of structural and crystallographic texture variations by acquiring complete 3D ED tilt series on a sample raster is time-consuming and challenging, as the electron dose would quickly reach prohibitive levels, depending on the targeted spatial resolution. By limiting the tilt series to only the most informative angles, both measurement time and beam damage can be drastically reduced, while spatial resolution can in turn be enhanced. For example, our previous study demonstrated that collecting ED patterns at only two tilt angles (0° for in-plane and

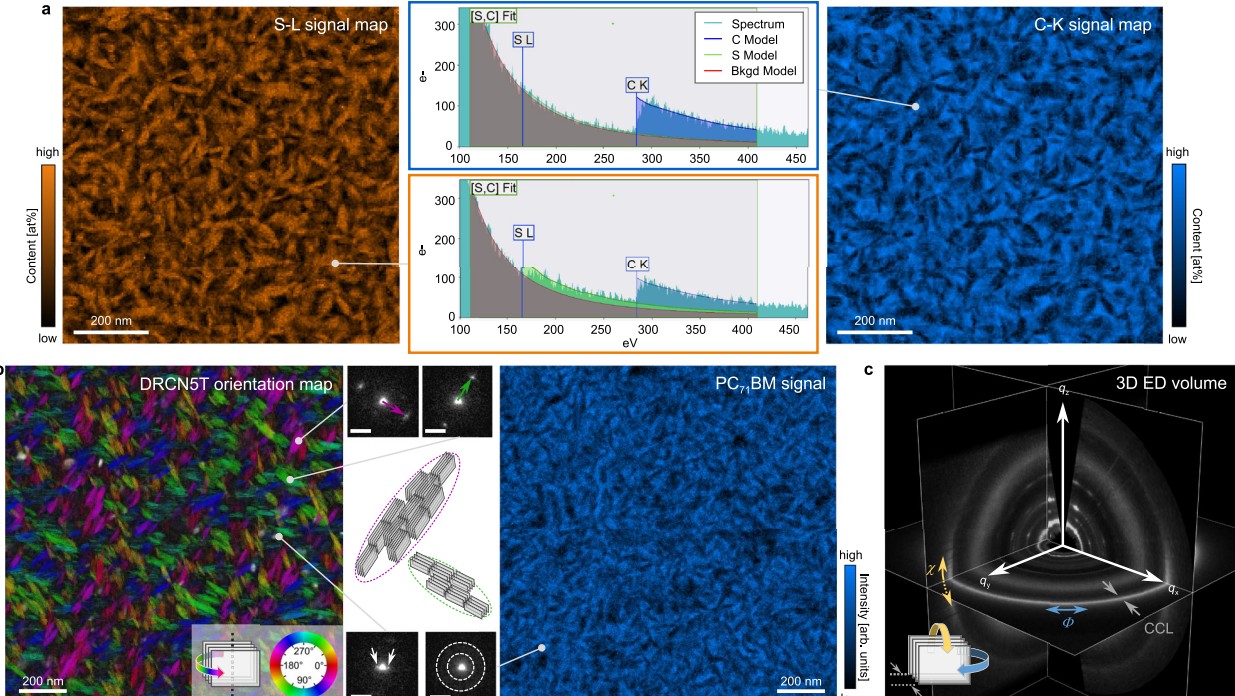

**Fig. 5 | Correlative TEM data. a** The STEM-EELS S-L map shows the leaf-like structure of the DRCN5T which contains more sulfur, whereas the C-K signal represents the carbon-richer PC$_{71}$BM. Exemplary single pixel spectra from different locations illustrate the observable signal and the applied model-based fitting approach. **b** 4D-SCED maps with exemplary underlying diffraction data (scale bars 2 nm$^{-1}$). The DRCN5T orientation map shows edge-on crystallites with color wheel representation based on in-plane π-π stacking peak position. A schematic arrangement of edge-on crystallites within a leaf-like domain based on the correlative TEM analysis is depicted as inset, highlighting the mosaic structure and π-π stacking direction along the short axis of the domain. The face-on crystallites as observable from (100) peaks close to direct beam are shown in gray scale. The PC$_{71}$BM distribution is mapped using a virtual ring detector covering the amorphous diffraction ring. **c** The ortho-plane view of the 3D ED volume highlights the true 3D character of diffraction information available from 3D ED, including crystallite in-plane-rotation ($\Phi$), out-of-plane-inclination ($\chi$) and CCL. This covers in-plane information not only within the $q_{xy}$ plane, but also along $q_z$. Inclined crystallites, differing from strict edge-on or face-on, are represented in the volume, as e.g., observable from $\chi$-width of the in-plane π-π stacking peak representing degree of alignment of edge-on crystallites.

~ 75° for a near out-of-plane view) provided the most essential structure information, which could be in principle spatially resolved if probed at different locations across the OSC film. Such a sparse sampling strategy enables mapping of regions of interest which are separated by micrometers, while keeping the electron dose per region low. In essence, 3D ED can be tailored to spatial resolution by trading off angular resolution—an approach particularly useful for investigating inhomogeneous samples. This compares favorably with GIWAXS, which typically averages over much larger areas (millimeter-scale), unless synchrotron micro-beam techniques are employed. Thus, while full 3D reciprocal-space mapping is dose-limited to an averaged region, targeted partial 3D ED scans can be performed on different locations to reveal spatial heterogeneities relevant to OSCs.

Furthermore, this 3D ED approach can be extended to other beam-sensitive photovoltaic materials, such as metal-halide perovskite films with sub-micron grains, provided certain conditions are met. First, the electron dose budget must remain within tolerance. Notably, many perovskites withstand cumulative doses of tens to hundreds of e$^-$ Å$^{-2}$ [62,63]—significantly higher than our model systems—which is favorable for 3D ED. Second, the film must be homogeneous over the few ×10$^3$ μm$^2$ area sampled by a full tilt series, unless inhomogeneities are the subject of study, as discussed above. In our experiments, each diffraction pattern was acquired from a ~ 3.4 μm region, with fresh areas illuminated for ~ 150 tilt increments (6 μm illumination diameter, spaced by 10 μm), corresponding to ~ 5000 μm$^2$ of sampled material area. We anticipate that a similar strategy can be applied to films with larger crystallite domains (hundreds of nm), particularly when using a larger selected-area aperture, which corresponds to probing a 10–20 μm diameter region of the sample and thereby ensures a statistically representative sampling volume at each tilt. Finally, crystalline materials with smaller unit cells (e.g., inorganic perovskites) scatter to higher angles. While the curved boundary of the missing wedge in GIWAXS along $q_z$ increases the likelihood of losing significant reflections, the straight missing wedge characteristic of 3D ED mitigates this effect.

Beyond these material inherent characteristics, sample preparation represents another key factor for extending the applicability of our 3D ED approach to other thin films. A practical prerequisite of the method is the use of electron-transparent, free-standing samples. In our study, this was achieved by detaching the OSC films from their substrate and transfer onto TEM grids. This step, however, may not be feasible for all material systems: fragile films are prone to cracking or warping during removal, while some may lack a convenient sacrificial layer (such as PEDOT:PSS) to facilitate release. In addition, the film must remain sufficiently flat on the TEM grid, as any curvature can distort the tilt geometry and complicate data interpretation (cf. Supplementary Fig. 11). Another prerequisite concerns sample thickness: the technique requires electron transparent regions typically in the range of ~ 100 nm, depending on the material. Moreover, it is important to account for the increase in projected thickness during tilting. For thicker films or multilayer, selective plan-view thinning or cross-sectioning is required. In this context, combining 3D ED with double-wedge preparation provides a possible route to depth-dependent analysis of structural and textural evolution across thicker films [64]. While originally developed for inorganic systems, this approach may also be extended to organic films. For cross-sectioning, FIB lift-out or

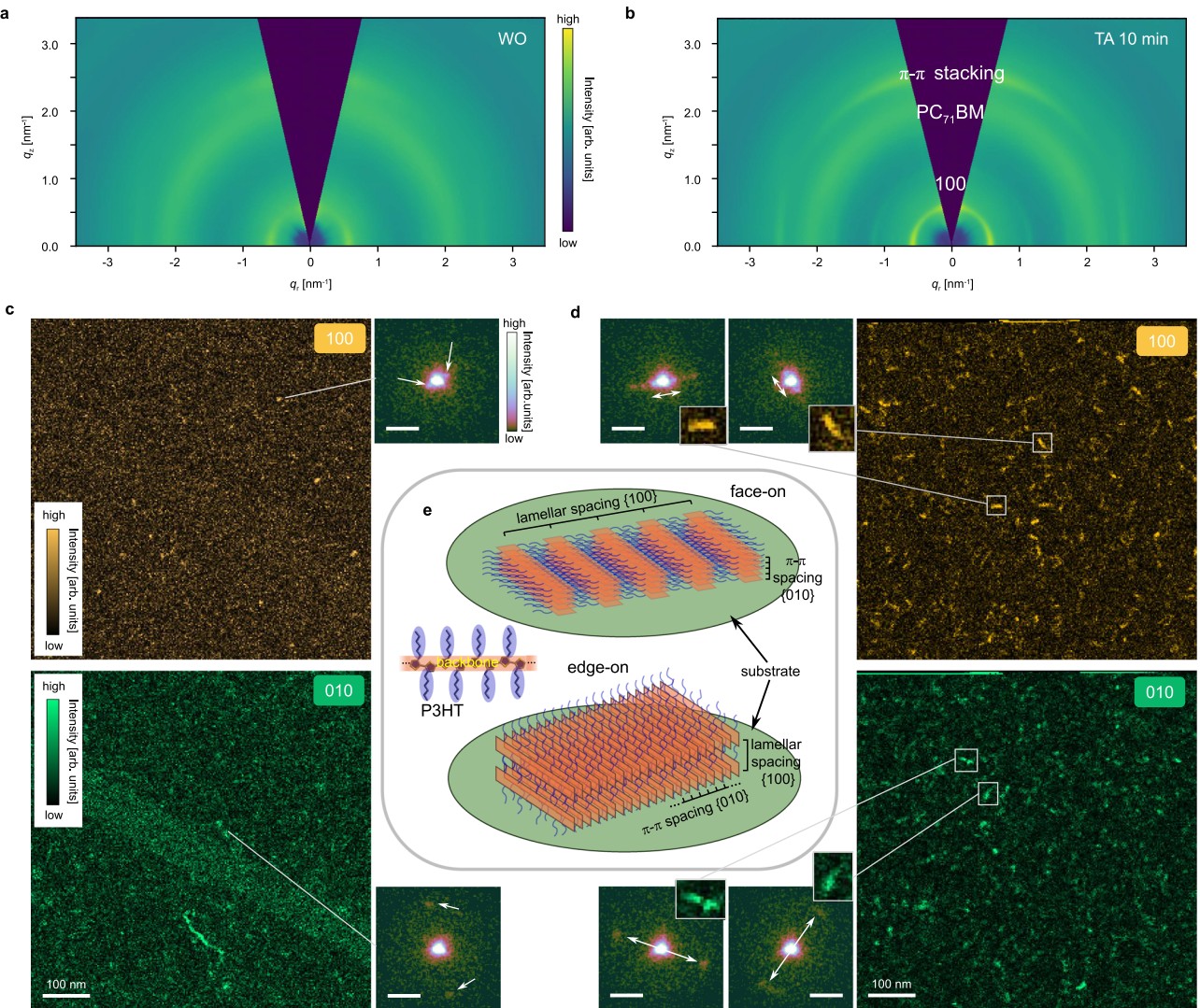

**Fig. 6 | Correlative analysis of structural evolution of P3HT:PC$_{71}$BM upon thermal annealing (TA). a, c** Results of untreated sample (WO); **b, d** results of 10 min TA sample. **a, b** 3D ED $q_{rz}$ maps. Upon TA the sharpened (100) ring shows intense in-plane and out-of-plane peaks, indicative of higher crystallinity with larger crystallite sizes in these two orientations. **c, d**, Face- and edge-on domains as revealed from virtual dark-field images from 4D-SCED by analysis of lamellar (100) and π-π stacking (010) diffraction signal respectively. Few raw diffraction patterns (intensity displayed in power scale) extracted from 2 × 2 probe pixel regions are

shown as insets (scale bars 2 nm$^{-1}$). The real space image area is magnified for one-to-one correspondence for the TA sample. **e** Scheme of molecular structure of P3HT and edge-on and face-on ordered regions. The P3HT structure was simplified to maintain clarity (backbone orange, side-chains blue). While the structure information in the substrate plane (green) can be directly observed from 4D-SCED analysis, the out-of-plane information can be deduced from CCLs analysis of 3D ED. For edge-on crystallites π-π spacing is in-plane, lamellar stacking is out-of-plane, for face-on crystallites vice versa.

microtoming can be employed. Care must be taken, however, to minimize process induced damage that could alter the native structure. We note these requirements to emphasize that, in its current implementation, our 3D ED approach is best suited for films that can either be fabricated directly on TEM-compatible membranes or transferred to them without significant perturbation.

Looking ahead, 3D ED will benefit from the progress in detector technology in electron microscopy[59,65,66]. Combined with automated data acquisition procedures and data science approaches[59], these developments will reduce the total acquisition time and required electron dose, enabling precise structural information to be obtained with higher throughput. Furthermore, advanced structural information retrieval routines and even structural refinement methods for polycrystalline and textured samples should be developed to enhance accuracy and broaden the applicability of 3D ED.

In summary, the results demonstrate that 3D ED enables investigation of the key structural characters like texture and mosaicity of beam-sensitive, polycrystalline organic thin films. Using solvent vapor

annealed DRCN5T:PC$_{71}$BM as model system, we described the complete workflow of 3D ED—from tilt series acquisition and analysis of diffraction patterns acquired at multiple tilt angles, to 3D reconstruction of reciprocal space and the extraction of $q_{rz}$ maps. We compared 3D ED with the well-established GIWAXS technique, highlighting differences in scattering geometry and reciprocal space sampling, discussing the origin and implication of the respective missing wedges. Laboratory and synchrotron GIWAXS measurements showed close agreement with 3D ED for the DRCN5T:PC$_{71}$BM film. We highlight the challenges and strengths of 3D ED and GIWAXS, demonstrating the potential of their correlative use. A central capability of 3D ED is the direct correlation of three-dimensional reciprocal space data with nanoscale structural, compositional, and crystallographic information acquired by imaging, nanoanalytical, and diffraction-imaging techniques within a single TEM instrument. Application of our method to a second OSC system (P3HT:PC$_{71}$BM), resolving post-processing induced structural changes, exemplifies that our approach can be extended to other organic, hybrid, and inorganic polycrystalline thin

films, where combined local and reciprocal-space analysis provides comprehensive information.

## Methods

### Sample preparation

The BHJ OSC samples were produced using a spin coating process. The active layers consisted of either the small molecule donor DRCN5T (2,2'-[(3,3''',3'''',4'-tetraoctyl[2,2':5',2'':5'',2''':5''',2''''-quinquethiophene]-5,5''''-diyl)bis[(Z)-methylidyne(3-ethyl-4-oxo-5,2-thiazolidinediylidene)]]bis-propanedinitrile, purity >= 99%, 1-Material, Dorval, Canada) and the fullerene acceptor $PC_{71}BM$ ([6,6]-Phenyl-C71-butyric acid methyl ester, purity >=99%, Solenne BV, Groningen Netherlands) (ratio 1:0.8 wt%), or the polymer donor P3HT (poly(3-hexylthiophene-2,5-diyl, average molecular weight 20,000–45,000, Sigma Aldrich Chemie GmbH, Taufkirchen, Germany) and the same $PC_{71}BM$ (ratio 1:1 wt.%). For the DRCN5T:$PC_{71}BM$ samples silicon wafer pieces of $1 \times 2$ cm were used as substrates due to the small surface roughness being beneficial for GIWAXS measurements (single side polished, terminated with $SiO_2$, Siegert Wafer, Aachen, Germany). The P3HT:$PC_{71}BM$ films were prepared on glass slides measuring $2.5 \times 2.5$ cm$^2$. The substrates were cleaned in acetone and isopropanol for 10 min each using an ultrasonic bath. On the dried substrates poly(3,4-ethylenedioxythiophene) polystyrene sulfonate (PEDOT:PSS, Clevios® P VP Al 4083, Heraeus, Hanau, Germany) in a mixture with isopropanol (ratio 1:4) was applied by doctor blading (amount of mixture 60 μl, 550 μm gap, blade speed 40 mm/s, plate temperature 50 °C). The active layer components were dissolved separately in chloroform before mixing the respective donor and acceptor solutions. The solutions were prepared under inert gas atmosphere and stirred at 40 °C with a speed of 150 rpm. The respective solutions were then mixed and further stirred before spin coating under inert gas atmosphere (1500 rpm for DRCN5T, 2000 rpm for P3HT as donor). As post-processing solvent vapor annealing was applied for the DRCN5T:$PC_{71}BM$ samples using a closed petri dish and 120 μl of carbon disulfide. For GIWAXS measurements the small molecule-fullerene samples were used, prepared as described so far. When post-processing was applied, the P3HT: $PC_{71}BM$ samples were thermally annealed for 10 min at 150 °C on a hotplate. For TEM the OSC thin films were detached from the substrates by immersing the substrate into a petri dish filled with distilled water. Thereby the PEDOT:PSS interlayer was dissolved and the active layer floated on the water surface. The active layer was then transferred to a Ni TEM support grid (200 mesh, lacey carbon grid).

### Characterization

TEM investigations were performed using either a double-corrected Titan Themis[3] 300 (ThermoFisher Scientific) equipped with a high-brightness field-emission gun (X-FEG) and a high-resolution post-column energy filter (GIF Quantum, Gatan, Inc., Pleasanton, USA), or a probe-corrected Spectra 200 (ThermoFisher Scientific) with an ultra-high-brightness cold field emission gun (X-CFEG) and a DECTRIS ARINA hybrid pixel detector. The Titan Themis was operated at 300 kV ($\lambda = 0.00197$ nm) for electron diffraction, (STEM-)EELS and EFTEM investigations. EF-SAED patterns were acquired with a nominal camera length of 145 mm and a 10 eV energy selecting slit around the zero-loss peak in a tilt series in 1° steps. The covered tilt angle range was −78° to 80° for the DRCN5T:$PC_{71}BM$ sample, −76° to 75° for the untreated P3HT:$PC_{71}BM$ and −75° to 75° for the thermally annealed P3HT:$PC_{71}BM$ sample. Zero-loss energy filtering is essential to suppress the strong inelastic scattering background at small scattering angles (which is crucial for detecting diffraction from molecular packing in -nm-scale distances). We have quantified the impact of the energy filter by comparing ED data collected with and without the 10 eV slit: as shown in Supplementary Fig. 12, filtering improves the signal-to-background and signal-to-noise ratio of low-$q$ diffraction features by effectively removing the diffuse background. Furthermore, we account for the

changing probed sample volume upon tilting when processing the intensities (see Supplementary Fig. 13 for details). Due to beam sensitivity of the samples each diffraction pattern was acquired from a fresh sample area (illumination diameter ~ 6 μm, contributing area ~ 3.4 μm via SAD aperture) using a beam flux of 0.45 e$^-$ Å$^{-2}$ s for 1 s. This corresponds to ~ 0.45 e$^-$ Å$^{-2}$ per pattern, well below the ~ 5 e$^-$ Å$^{-2}$ critical dose for maintaining crystallinity. The samples were translated ~ 10 μm between acquisitions to avoid cumulative damage, resulting in ~ 5000 μm$^2$ sampled area for tilt series acquisition. The beam damage series of EF-SAED patterns of the P3HT:$PC_{71}BM$ samples for determining critical dose was acquired at one position with a frame integration time of 50 ms. The STEM-EELS elemental mapping of the DRCN5T:$PC_{71}BM$ sample was performed after completing the ED tilt series: a 100 pA probe with 5 ms dwell and 4.5 nm step yielded a dose rate of ~ $3.08 \times 10^9$ e$^-$ Å$^{-2}$ s, summing to 15,407 e$^-$ Å$^{-2}$ instantaneously (i.e., per scan point) or 94 e$^-$ Å$^{-2}$ over the full $200 \times 200$ scan frame, meaning that long-range crystalline order was already destroyed during EELS. However elemental composition (particularly for heavier elements like S) remains as long as the elements are not physically removed from the sample (no mass loss below the evaporation threshold), or diffused. EFTEM images of the P3HT:$PC_{71}BM$ samples were acquired for the $PC_{71}BM$ plasmon peak at 30 eV using a slit width of 5 eV and an exposure time of 5 s. The Spectra 200 was operated at 200 kV for 4D-SCED acquisition. A probe current of 21 pA, probe convergence semi-angle 0.85 mrad, sample defocus of 4 μm, step size of 2.7 nm and dwell time of 40 μs was used for the DRCN5T:$PC_{71}BM$ sample, which corresponds to an instantaneous fluence of 1.4 e$^-$ Å$^{-2}$ and frame fluence of 7.0 e$^-$ Å$^{-2}$ (due to oversampling). For the P3HT:$PC_{71}BM$ samples, the probe current was reduced to 14 pA while keeping other conditions the same. In both cases, scanning pixels were set to $512 \times 512$ and a quarter field of view cropped out for visualization.

The laboratory GIWAXS datasets were acquired at the versatile advanced X-ray scattering instrument Erlangen (VAXSTER) which is equipped with a liquid metal-jet Ga source (Excillum, Kista, Sweden; Ga-K$_\alpha$ $\lambda = 1.341$ Å) and four collimating, four-bladed slits. The Pilatus 300 K detector consists of 3 modules, resulting in two stripes of missing information in the detected patterns. This can be bypassed by summing two measurements acquired with shifted detector positions (Supplementary Fig. 14). A sample detector distance (SDD) of 178.7 mm was determined via calibration with a AgBh sample and an incidence angle of ~ 0.14° was calculated using the calibrated SDD in combination with direct and reflected beam positions. This incidence angle is approximately the critical angle of the active layer and below the critical angle of the Si substrate ($\alpha_{c,Si} = 0.19°$). The instrument was operated under vacuum conditions and the illumination time was 16 h for each pattern. The direct beam position was extracted out of a measurement with the sample removed from the beam path, for determination of the reflected beam position an additional, shorter measurement of 60 s was conducted as in the longer 16 h measurements the reflected beam pixels are saturated, which makes peak position determination challenging.

The synchrotron GIWAXS measurement was acquired at the High Resolution Diffraction Beamline P08, PETRA III, DESY[67]. The X-ray energy was 25 keV with a beam size of $0.4 \times 0.1$ mm, a measurement time of 10 s and an incidence angle of 0.072°. A XRD 1621 flat panel detector (Perkin Elmer Inc., Waltham, MA, USA) was used with a SDD of 704 mm.

### Data processing and visualization

The electron diffraction and STEM-EELS data were evaluated using Gatan Microscopy Suite (GMS3) software with public plugins and home-developed scripts. The STEM-EELS maps were computed with model-based quantification method as implemented in GMS3[47].

A home-made processing workflow[68] was implemented to reconstruct the three-dimensional reciprocal space volumes, based

on which the $q_{rz}$ maps were calculated, for gaining insights into 3D structure and quantitative comparison with GIWAXS. The raw diffraction data was pre-processed involving the following steps: center determination, tilt axis identification, diffraction pattern rotation and intensity normalization. The diffraction pattern centers were used to determine and correct pattern shifts before further processing. The tilt axis was identified from the instrumental setup and the symmetry of diffraction patterns at the highest tilt angles. Each pattern was then rotated so that the tilt axis coincided with the vertical image ($q_y$-)axis. To account for different projected sample thickness and interaction volume, the intensity of each diffraction pattern was normalized to the sum intensity of the presumably isotropic $PC_{71}BM$ diffraction halo ring of the respective diffraction pattern (cf. Supplementary Fig. 13).

Afterwards, the three-dimensional reciprocal space volume was reconstructed based on a polar transformation approach, detailed as follows. Each diffraction pattern is a slice of reciprocal space defined as in the $q_{xyz}$ cube where $q_z$-axis and $q_y$-axis coincide with the electron traveling direction and the experimental tilt axis, respectively. Every diffraction pattern in the aligned data stack can be viewed as having a common $q_y$-axis, and their $q_x$-axes in the patterns lie in the $q_{xz}$ plane and are rotated by angle $\alpha$ (with $\alpha$ being zero coincident with $q_x$-axis and positive $\alpha$ in anti-clock wise direction). In the tilt series, each diffraction pattern is a slice sampling the three-dimensional reciprocal space (see Fig. 2) with the density of sampling closer to the tilt axis higher than away from the tilt axis. We first create a new orthogonal data volume expressed in cylindrical coordinates $q_x$-$\alpha$-$q_y$ and then fill the aligned raw data into this new volume. The positive side (along $q_x$-axis) of each diffraction pattern fills the $q_{x'y}$ plane at $\alpha$ position along the $\alpha$-axis, while the negative side of each diffraction pattern is filled in a plane at $\alpha$ position shifted by 180 degrees (along $\alpha$ axis). Finally, all the $q_x$-$\alpha$ planes along the $q_y$-axis are transformed back to $q_{xz}$ plane by plane according to polar transformation. In such way, we obtain the three-dimensional reciprocal space volume not only computationally efficient, but also the transformation implies an interpolation of the slice sampling at high distance to the tilt axis. The different sampling density at various distance to the tilt axis would result in increased diffraction intensities closer to the tilt axis in the reconstructed volume, which is normalized by a unit data volume of the same shape as the aligned diffraction data and reconstructed via the same scheme. In this way, the three-dimensional reciprocal space volume is reconstructed in which the scattering volume (due to thickness and projection effect) effect and sampling multiplicity are considered, whereby the diffracted intensity can be further quantitatively evaluated. This method works very robust for datasets acquired at equal tilt increment in the experiment.

A $q_{rz}$ map can be generated by azimuthally integrating the $q_{xy}$ planes along $q_z$-axis by polar transformation of the $q_{xy}$ planes and projection along the angle axis (see Fig. 3b). The Friedel-pairs of diffraction intensities in the negative side along the $q_z$-axis are expected to be the same with the positive side. Therefore, we flip the negative side average with the positive side to further improve SNR. In this way, the $q_{rz}$ map, similar to that of GIWAXS, is generated for quantitative comparison.

The three-dimensional reciprocal space volume was visualized using the software ChimeraX with appropriate viewing modes[69]. The ortho-plane view displayed in Fig. 3b consists of 3 orthogonal planes of the volume with $q_{xyz}$ as normal vectors of the planes. The videos of the volume in SI utilize a maximum intensity projection.

$\chi$-linecuts were generated out of the 3D ED $q_{rz}$ map by choosing ring segments of specified radial range, polar transformation and subsequent projection along the radius range. Extraction of linecuts was done using the whole available $q_{rz}$ map information, i.e., not only one of the before mentioned data pads of negative and positive $q_{rz}$

average. Generation of the $q_{rz}$ map version with applied scale and color look-up-table was achieved using ImageJ[70] and a Python script based on Matplotlib[71].

For the laboratory GIWAXS data calibration of the SDD and determination of positions of direct and reflected beams by two-dimensional Gaussian fits, the software Fit2D was used[72]. Furthermore, the data were evaluated with the MATLAB based software GIXSGUI. This includes transformation into reciprocal space coordinates and extraction of different linecuts[73]. The summing of the two GIWAXS measurements with respect to the different detector positions was done using ImageJ. Peak fitting for all data shown was achieved using LIPRAS software[74].

The synchrotron GIWAXS data were evaluated using Python-based Jupyter Notebooks[75], based on scripts provided by H.G.S. and F.B. Processing included reciprocal space transformation and linecut extraction. Main python packages used were pygix, pyFAI[76], FabIO[77], Matplotlib[71] and NumPy[78].

The 4D-SCED data were analyzed and visualized using the routines published in our earlier work[40].

## Data availability
Electron microscopy and GIWAXS data generated in this study are available in the open-access repository ZENODO under the accession code 10.5281/zenodo.18484796[79].

## Code availability
The code used for 3D ED data processing is available in the open-access repository ZENODO under the accession code 10.5281/zenodo.18484796[79]. This includes a README file with step-by-step instructions to facilitate use of the code by the community.

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

## Acknowledgements

The authors gratefully acknowledge funding by the German Research Foundation (DFG) through the Collaborative Research Center "ChemPrint" (project 538767711, CRC 1719) and the Research Training Group "CorMic" (project 537140136, RTG 3103). We acknowledge DESY (Hamburg, Germany), a member of the Helmholtz Association HGF, for the provision of experimental facilities. Parts of this research were carried out at Petra III using P08 beamline. Beamtime was allocated for proposal R-20240703., M.H. and H.G.S. acknowledge funding from the German Federal Ministry of Research, Technology and Space (BMFTR) via project 05K22PP1 and 05K24CJ1. M.H., A.K., and H.G.S. acknowledge the funding by the German Federal Ministry of Research, Technology and Space (BMFTR) and the Ministry of Economic Affairs, Industry, Climate Action and Energy of the State of North Rhine-Westphalia through the project HC-H2. A.K. thanks the Stiftung Stipendien-Fonds of the German Chemical Industry Association (Verband der Chemischen Industrie, VCI) for a Kekulé fellowship.

## Author contributions

I.K.: writing—original draft, writing—review & editing, software, formal analysis, investigation, visualization. M.W.: writing—original draft, writing—review & editing, methodology, software, formal analysis, investigation, visualization, supervision. S.R.: writing—review & editing, investigation, supervision. J.W.: writing—original draft, writing – review & editing, supervision, formal analysis. S.M.: investigation. K.D.: investigation, formal analysis. A.K., M.H.: Software. L.L.: resources. F.B.: investigation, resources. H.G.S.: writing—review & editing, software, resources. T.U.: resources. C.J.B.: resources. E.S.: writing—original draft, writing—review & editing, conceptualization, methodology, resources, supervision, project administration, funding acquisition.

## Funding

## Competing interests

The authors declare no competing interests.
