## [Transparent Peer Review file · Nature Communications]

3D Electron Diffraction – The Missing Slice Completing Nanoscale Analysis of Organic Solar Cells in TEM

Corresponding Author: Professor Erdmann Spiecker

Version 0:

Reviewer comments:

Reviewer #1

(Remarks to the Author)

In the manuscript titled "3D Electron Diffraction – The Missing Slice for a Complete Nanoscale Analysis of Organic Solar Cells in TEM", Kraus et al. present an elastically filtered 3D Electron Diffraction (3D ED) approach capable of resolving the crucial structural parameters of organic thin films. They provide a thorough explanation of the experimental setup, data processing, and underlying working mechanism, all of which will be illuminating for seasoned experts and newcomers alike. Furthermore, they demonstrate that 3D ED yields results comparable to GIWAXS, suggesting an expanded role for TEM in materials research. While the work is both interesting and impressive, further clarification of its novelty is necessary, especially regarding what sets 3D ED apart from existing methods. In addition, questions remain about the technique's applicability to a broader range of materials, including polycrystals with larger grain sizes.

Below, I detail my comments:

1. As the author mentioned on page 9, 'to mitigate beam damage, each pattern was acquired on unexposed area via systematic shifting the sample', and the total probed area in 3D ED is about 1000 μm^2 range, each illuminated area was 6 μm in diameter. My main concern is that this acquisition mode presupposes microscale homogeneity in the sample to ensure reconstruction consistency. Otherwise, diffraction patterns may differ not only by tilt angle but also by sample position, potentially causing data-processing failures. Hence, this method can align well with GIWAXS only for certain types of samples. Can the authors extend this approach to other beam-sensitive photovoltaic materials such as polycrystalline halide perovskite films with grain sizes in the hundreds of nanometers? While this 3D ED still maintaining reliable reconstructions?
2. One primary advantage of 3D ED over GIWAXS is its compatibility with various TEM characterization techniques (e.g., imaging, spectroscopy, and 4D-STEM). However, the manuscript does not present real-space bright-field and dark-field images of the DRCN5T:PC71BM film to illustrate its morphology and grain structure. Furthermore, to effectively correlate 3D ED data with real-space observations, it would be helpful to indicate on the real-space images precisely where 3D ED measurements were taken and the region used for ED acquisition.
3. The manuscript states that "elastic filtering is essential to suppress the pronounced inelastic scattering background at small scattering angles," and all the presented results are energy filtered. To underscore the importance of this filtering, the authors should compare 3D ED measurements obtained with and without the 10 eV energy-selecting slit.
4. The reported 130 pA beam current for EELS acquisition seems high for such beam-sensitive materials. What were the electron dose rate and total dose during EELS measurements? Moreover, although the S L-edge map in Figure S8b exhibits a satisfactory signal-to-noise ratio, comparable to that of the C K-edge map, the S L-edge is barely visible in Figure S8c. It would be helpful for the authors to explain how they produced the S L-edge map despite its apparently low signal, particularly given the strong C K-edge. Additionally, the oxygen K-edge and nitrogen K-edge features are undetectable in the spectrum, raising further questions about measurement conditions and processing.

(Remarks on code availability)

Reviewer #2

(Remarks to the Author)

The authors report the use of 3D electron diffraction (3D ED) based on TEM to analyze the ordered structure of a bulk-heterojunction (BHJ) film, achieving analytical precision comparable to that of GIWAXS. The authors highlight the primary

advantage of 3D ED as enabling the combination of molecular packing and morphological characteristics within the BHJ film. Nevertheless, the manuscript did not clearly discuss the specific correlation between these morphological features and molecular packing and explain why this point is significant. While 3D ED has proven to be a powerful tool for analyzing the structures of microcrystalline particles, e.g., MOFs or COFs, whose single crystals are difficult to obtain, this reviewer believes it cannot replace GIWAXS for film structure analysis. This is due to several limitations on ED: 1) The testing area of ED is only scaled in nanometer, whereas the GIWAXS beam spot can be tens of micrometers; 2) ED cannot probe directions close to the in-plane direction (GIWAXS uses incident angles smaller than 1 degree); 3) The requirements for in situ ED studies are more stringent (GIWAXS can be readily integrated with various stages, such as heating/cooling stages and optical microscopes). More importantly, X-ray scattering has long enabled the collection of clear diffraction signals in both horizontal and vertical directions from large-area organic thin-film samples and has successfully resolved their 3D structures (Science 2010, 330, 808; Science 2015, 348, 1122). Therefore, while acknowledging that this paper represents the first trial of 3D ED to BHJ film structure analysis, the reviewer concludes that, compared to the well-established GIWAXS technique, it does not constitute a breakthrough or demonstrate convincing advantages. Furthermore, the manuscript fails to demonstrate the broader applicability of 3D ED to different organic solar cell systems or its scientific significance through structure-property relationship studies. Finally, the first eight pages devote excessive space to background information together with Figures 1 and 2, while the experimental data in the manuscript as well as supporting data (Figs. S1-S10) appear too weak and thus fall short of the standard for a research paper. In light of the above comments, the paper would not be recommended for publication in Nature Communications.

(Remarks on code availability)

None

Reviewer #3

(Remarks to the Author)

The work by Irene Kraus et al. discusses the use of 3D electron diffraction for the characterization of organic thin films with special attention to how this compares to X-ray based grazing incidence wide angle scattering. The results are quite impressive and the comparisons to GIWAXS are both very clear and very convincing. The authors provide an appropriate level of detail for others to adopt this approach and I expect that it will find a welcome audience in the field of organic solar cells. The manuscript is well written and clearly presented, I recommend acceptance after only minor revisions.

From reading the manuscript, the main advantage of the 3D ED approach (especially compared to GIWAXS) seems to be the ability to combine it with other TEM based characterization, such as imaging or EELS. Given the need to translate the sample between ED images to avoid beam damage, the finite size of the electron spot, and the changing footprint during rotation, additional discussion of the feasibility of spatially resolved 3D ED would be welcome. Does this present a fundamental limit to the size of different regions which can be measured in a film, and how does this compare to the features which might be of interest especially for OSC's?

The film was removed from the substrate for 3D ED measurements, with this in mind, a more detailed discussion of the sample limitations for this technique seems appropriate. This is briefly touched on in the discussion section.

(Remarks on code availability)

While I have briefly looked through the code I am not expert enough to evaluate it. However, a README with instructions for how to get started with using this should be included.

Version 1:

Reviewer comments:

Reviewer #1

(Remarks to the Author)

The authors have fully addressed my previous concerns, and high-quality data has been supplemented. I suggest acceptance of this work for publication in Nature Communications.

(Remarks on code availability)

Reviewer #2

(Remarks to the Author)

The authors have updated their manuscript by providing additional experimental results to demonstrate the generality of 3D ED in another organic solar cell (OSC) system (P3HT:PCBM), as well as its correlation with other TEM techniques, including STEM-EELS and 4D-SCED, to elucidate the relationship between the morphology and molecular packing. While these enhancements improve the technical quality and scientific rigor of the paper, the revision has only partially addressed

my concerns and does not sufficiently strengthen its suitability for Nature Communications based on the following comments:

1) The 3D ED method is not a novel characterization technique for determining crystal structures of nanomaterials where single crystals are unavailable. Its application in the OSC field. For instance, the identification of molecular packing in P3HT by 3D ED has been well established for over a decade, as evidenced by the pioneer works of Martin Brinkmann and co-workers (e.g., *Macromolecules* 2007, 40, 7532; *Macromolecules* 2010, 43, 4961; *Adv. Funct. Mater.* 2016, 26, 408; *Mater. Chem. Front.* 2020, 4, 1916). In light of their studies, the generality of 3D ED that the authors aim to demonstrate in this work appears less meaningful.

2) While I acknowledge that GIWAXS has directional limitations and cannot provide full crystallographic information, 3D ED similarly fails to deliver 3D real-space nanomorphology in OSC studies. For example, Figure 4 only illustrates a structural transition from random orientation (diffraction rings) to anisotropic orientation (diffraction arcs) via tilt angle adjustment, without clarifying how the small molecules pack in 3D or in thin films. No illustrative model of molecular arrangement is provided neither. Moreover, confirming edge-on or face-on configurations via 3D ED, which is as also effectively accomplished by GIWAXS in Figure 5, does not offer substantial additional insight.

3) The writing and organization of the manuscript remain exceedingly poor. At times, it reads more like a review article than a research paper. The discussion relies heavily on common knowledge, which are readily accessible via Google or general literature, rather than on critical analysis of the experimental results. Furthermore, the manuscript is unnecessarily lengthy, yet contains limited substantive data, or original insight. Both in content and structure, it falls short of the standards expected of a high-quality research publication.

4) The manuscript merely posits that 3D ED can be used as a tool for studying OSCs, without justifying why this method is necessary or what novel insights it can provide. Key questions remain unaddressed: What unprecedented discoveries does it enable in OSC research? How can it reveal new mechanisms, enhance material performance, or provide design principle of OSC materials? Without discussing these innovative aspects, the paper is unlikely to attract broad interest from the readership.

In summary, although the experimental techniques and conclusions are technically sound, the revised manuscript fails to overcome the fundamental concerns regarding novelty and structure that I raised in the initial review. Finally, as to whether this manuscript merits publication in Nature Communications, I respect and defer to the editor's judgment.

(Remarks on code availability)

None

Reviewer #3

(Remarks to the Author)

The authors have done a thorough job of addressing my comments as well as those of the other reviewers. I have no additional concerns which should be addressed before publication.

(Remarks on code availability)

Version 2:

Reviewer comments:

Reviewer #1

(Remarks to the Author)

The authors have addressed all my concerns. The supplemented data have further strengthened the technical merit of this work, and the clarity has also been improved by reorganizing figures and descriptions. From the perspective of an electron microscopy scientist, this work has a high degree of completion. I recommend the immediate acceptance of this manuscript.

(Remarks on code availability)

Reviewer #2

(Remarks to the Author)

I sincerely appreciate the authors' efforts in addressing my comments, which have led to great improvements in the revised manuscript. While the content has been strengthened, the writing and overall organization of this paper are not yet satisfactory. Regarding the final decision on this revision, I thus defer to the editor's judgment.

(Remarks on code availability)

None

Response to Reviewer Comments

Dear Editor and Reviewers,

We thank you for the opportunity to revise our manuscript “**3D Electron Diffraction – The Missing Slice for a Complete Nanoscale Analysis of Organic Solar Cells in TEM.**” We are grateful to the reviewers for their thorough evaluation and constructive feedback. In response, we have carefully revised the manuscript and Supplementary Information in accordance with the comments. Below, we provide a point-by-point response to each comment. For clarity, the reviewers’ comments are italicized (with quotation marks for direct excerpts), and our responses follow in regular font. All changes made to the manuscript text are again italicized. We also indicate revisions in the manuscript by referring to page and line numbers (from the color highlighted version) and, where appropriate, quoting the original and revised text. We have endeavored to address all concerns fully, both through modifications to the text (highlighted in the revised manuscript) and, where relevant, additional data or explanation.

We believe these revisions significantly strengthen the manuscript and we hope that the changes will satisfy the reviewers’ expectations. Detailed responses are given below.

Reviewer #1 (Comments and Responses)

We thank Reviewer #1 for the positive evaluation of our work and the insightful suggestions. We have addressed all of the points raised, as detailed below.

1. Comment (Reviewer #1): *“As the author mentioned on page 9, ‘to mitigate beam damage, each pattern was acquired on unexposed area via systematically shifting the sample’, and the total probed area in 3D ED is about 1000 μm^2 (each illuminated area $\sim 6 \mu\text{m}$ in diameter). My main concern is that this acquisition mode presupposes microscale homogeneity in the sample to ensure reconstruction consistency. Otherwise, diffraction patterns may differ not only by tilt angle but also by sample position, potentially causing data-processing failures. Hence, this method can align well with GIWAXS only for certain types of samples. Can the authors extend this approach to other beam-sensitive photovoltaic materials such as polycrystalline halide perovskite films with grain sizes in the hundreds of nanometers? While this 3D ED still maintaining reliable reconstructions?”*

Response: We appreciate the reviewer’s concern regarding sample homogeneity and the applicability of our 3D ED approach to other materials (e.g., large-grained perovskite thin films). In general, we agree that successful 3D ED requires the sample to be relatively homogeneous over the total probed area, so that each diffraction pattern in the tilt series is representative of the same overall structure. We have now added text to the manuscript explicitly discussing the conditions under which 3D ED can be extended to other beam-sensitive systems and still yield reliable reconstructions. In particular, we note that:

- **Beam damage considerations:** Many hybrid perovskite films can tolerate higher electron doses (on the order of 10–100 $\text{e}^-/\text{\AA}^2$) than the organic system studied in our work, which suggests that 3D ED is even more feasible on such materials. We now state in the Discussion that the critical dose of typical metal-halide perovskites is higher than that of our model system, providing a larger dose budget for 3D ED data acquisition (page 21, lines 453-457).

- **Sample area and homogeneity:** To ensure consistency across tilts, the sample should be homogeneous over an area of at least a few thousand square microns (comparable to the total area probed in a full tilt series). In our experiment, each 3D ED pattern was collected from a ~ 3.4 μm diameter region (defined by the selected area aperture) and we used ~ 150 distinct regions for a full tilt series (with an ~ 10 μm translation between each). We have clarified in the revised text that this method effectively samples ~ 5000 μm^2 of the film, which was achievable because the DRCN5T:PC₇₁BM blend was uniform at that scale. While large-scale homogeneity is generally a prerequisite for absorber layers, we would like to emphasize that the relatively local nature of the 3D measurement, compared to GIWAXS, provides an important opportunity to identify and analyze inhomogeneities in the film texture that might otherwise be overlooked in GIWAXS (for further details on the potential of spatial resolution for investigating inhomogeneity, see Reviewer #3, Comment 1).
- **Grain size consideration:** To obtain robust statistics in 3D ED, a sufficient number of grains must be included in the sample area defined by the selected-area aperture. In this context, we note that modern TEMs enable the use of larger selected-area apertures (corresponding to electron probe areas up to ~ 10 – 20 μm in diameter) without compromising beam parallelism and diffraction pattern quality. Thus, even films with grains on the order of a few hundred nanometers can be accommodated by averaging over a larger area per tilt, if needed. This point is now emphasized on page 21 (lines 460-463) of the revised manuscript.
- **Unit cell and missing wedge consideration:** We have also added a remark that materials with smaller unit cells (such as inorganic perovskites) diffract at larger scattering angles. As a result, reflections with out-of-plane components are more likely to be missed in GIWAXS due to the curved missing wedge imposed by the smaller size of the Ewald sphere. In contrast, 3D ED—with its straight missing wedge arising from the limited tilt angle range— offers a clear advantage in this respect. (page 21, lines 463-466).

In summary, we believe the 3D ED approach can indeed be extended to other beam-sensitive thin-film materials, provided that the film is sufficiently uniform over the probed area, flat enough to maintain consistent tilt geometry, and that the electron dose is managed to stay below the damage threshold. To demonstrate broader applicability, we have also expanded our study to include a second organic solar cell system (the well-known P3HT:PC₇₁BM blend) in the revised manuscript. This additional dataset (described in our response to Reviewer #2 Comment 3 below) will show that 3D ED can resolve subtle structural changes upon thermal annealing in a different OSC material, in line with what has been observed by GIWAXS for that system. We have added a subsection in the manuscript indicating the new P3HT:PC₇₁BM results (see pages 16-17, lines 337-355) – this is intended to reassure the reviewers that data from another material system are forthcoming to support our claims of generality.

Changes in Manuscript (Comment 1): We have added a new paragraph in the Discussion section addressing the extension of 3D ED to other systems. For example, we inserted the following text on page 21:

- Original text: (no explicit discussion of applicability to other materials was given)
- Revised text (page 21, lines 453-466): *“Furthermore, this 3D ED approach can be extended to other beam-sensitive photovoltaic materials, such as metal-halide perovskite films with sub-micron grains, provided certain conditions are met. First, the electron dose budget must remain within tolerance. Notably, many perovskites*

withstand cumulative doses of tens to hundreds of $e^-/\text{\AA}^2$ ^{61,62} - significantly higher than our model systems – which is favorable for 3D ED. Second, the film must be homogeneous over the few $\times 10^3 \mu\text{m}^2$ area sampled by a full tilt series, unless inhomogeneities are the subject of study, as discussed above. In our experiments, each diffraction pattern was acquired from a $\sim 3.4 \mu\text{m}$ region, with fresh areas illuminated for ~ 150 tilt increments ($6 \mu\text{m}$ illumination diameter, spaced by $10 \mu\text{m}$), corresponding to $\sim 5000 \mu\text{m}^2$ of sampled material. We anticipate that a similar strategy can be applied to films with larger crystallite domains (hundreds of nm), particularly when using a larger selected-area aperture, which corresponds to probing a $10\text{--}20 \mu\text{m}$ diameter region of the sample and thereby ensures a statistically representative sampling volume at each tilt. Finally, crystalline materials with smaller unit cells (e.g., inorganic perovskites) scatter to higher scattering angles. While the curved boundary of the GIWAXS missing wedge along q_z increases the likelihood of losing significant reflections, the straight missing wedge characteristic of 3D ED mitigates this effect.”

2. Comment (Reviewer #1): *“One primary advantage of 3D ED over GIWAXS is its compatibility with various TEM characterization techniques (e.g., imaging, spectroscopy, and 4D-STEM). However, the manuscript does not present real-space bright-field and dark-field images of the DRCN5T:PC71BM film to illustrate its morphology and grain structure. Furthermore, to effectively correlate 3D ED data with real-space observations, it would be helpful to indicate on the real-space images precisely where 3D ED measurements were taken and the region used for ED acquisition.”*

Response: We fully agree with the reviewer that correlating the diffraction data with real-space morphology is of great importance, and we apologize for not having provided sufficiently clear examples of these correlations in the original submission. In fact, we did acquire real-space images and maps of the DRCN5T:PC₇₁BM film – these were partially shown in Figure 1 of the original manuscript, but not sufficiently emphasized. Figure 1 (in the initial submission) included a STEM image and a STEM-EELS elemental map of the active layer, which reveal the nano-morphology of the phase-separated blend. Additionally, we have 4D-STEM data (acquired with 4D-Scanning Confocal Electron Diffraction, 4D-SCED approach {Wu, M. et al., Nature Communications 2022, 13, 2911}) from the identical sample, which provides local crystallographic orientation and packing information with respect to the morphology. However, we acknowledge that the presentation of Figure 1 focused more on the concept of the “missing slice” in TEM characterization of organic absorber layers and the need for 3D ED, rather than explicitly linking those real-space images to the diffraction results.

In the revision, we have not only modified Figure 1 and its description to strengthen the real-space correlation aspect, but also added a separate subsection in the results section dedicated to correlative TEM analysis (described in response to Reviewer #2, Comment 1). Specifically, in Figure 1 we have enlarged and more clearly shown the correlative TEM datasets of the DRCN5T:PC₇₁BM film corresponding to the respective methods. We also adjusted the figure caption and the main text to explain that these images are actual data from the same sample used for 3D ED, highlighting the compatibility of 3D ED with real-space imaging and spectroscopy on the identical sample. The additional subsection in the results section, includes a new Figure 6 showing detailed analysis and underlying raw data for the STEM-EELS and 4D-SCED datasets from Figure 1, supporting the comprehensive correlative TEM analysis in combination with 3D ED (described in response to Reviewer #2, Comment 1).

In summary, in the revised manuscript text, we now explicitly describe the content of Figure 1 as a correlative dataset. In context of a dedicated new subsection in results section with an additional Figure 6 we highlight the correlative approach of our method: the ADF-STEM image and elemental (STEM-EELS) maps show the blended morphology (with donor- and acceptor-rich phases), and the 4D-STEM/SCED analysis (presented schematically in Figure 1 and in detail in the new Figure 6) reveals crystalline domains and their orientation within that morphology.

These additions underscore the primary advantage of our approach: because all measurements (imaging, diffraction, EELS mapping) are done in the TEM on the same sample, we can directly correlate molecular packing (from 3D ED and 4D-STEM) with nanoscale morphology and composition (from TEM imaging and STEM-EELS mapping). This point has now been made more explicit. We believe this clarification strengthens the manuscript by illustrating why combining 3D ED with real-space techniques is so powerful.

Changes in Manuscript (Comment 2): The revisions include changes to the text on pages 4-5 of the main manuscript, and to Figure 1 and its caption:

- **Original text** (pages 4-5, lines 102–106): “... (cf. Figure 1) Using an extensively studied model BHJ system (DRCN5T:PC₇₁BM)^{8,15,16,37,38} we develop a 3D ED workflow and validate its reliability by comparing results with GIWAXS from laboratory and synchrotron sources. Our results demonstrate that 3D ED provides critical structural information, revealing molecular texture and mosaicity⁵ inaccessible to conventional techniques.”
- **Revised text** (page 5, lines 105–112): “... (cf. Figure 1). Using a well-studied model BHJ system (DRCN5T:PC₇₁BM)^{8,15,16,39,40}, we develop a 3D ED workflow and validate its reliability by comparing results with GIWAXS from laboratory and synchrotron sources. By integrating 3D ED with TEM-based imaging, diffraction, and analytical capabilities, we establish it as a powerful complementary method to GIWAXS for the comprehensive characterization of OSCs. Using the same model BHJ sample, we illustrate the range of multi-modal data obtainable from a single OSC specimen in the TEM (cf. acquired data in Figure 1). Our results demonstrate how detailed molecular packing information from 3D ED, revealing molecular texture and mosaicity, can be directly correlated with these morphological features.”
- **Figure 1 Caption:** We have rewritten the Figure 1 caption to describe the real-space images and their link to 3D ED: “Figure 1. 3D ED completing the structural characterization of organic semiconductor thin films in TEM. The central scheme illustrates the integration of 3D ED into the suite of analytical, imaging, diffraction and diffraction imaging methods offered by a single TEM instrument. The addition of 3D ED provides insights into 3D texture and mosaicity, complementing already available information on chemical composition, nanomorphology, in-plane molecule ordering, and local molecule packing, thereby significantly enhancing the characterization capabilities of OSC thin films. Surrounding the scheme, the outer ring of images presents data measured on a single sample of the DRCN5T:PC₇₁BM OSC model system in the TEM, aligned with the corresponding TEM methods. This comprehensive characterization facilitates the direct correlation of real and reciprocal space information, linking 3D molecular packing with nanoscale morphology and composition.”

By implementing these changes, we make it clear in the paper that we have real-space images of the film's morphology and that we use them to correlate and pinpoint where diffraction data were obtained, directly addressing the reviewer's recommendation.

3. Comment (Reviewer #1): *“The manuscript states that ‘elastic filtering is essential to suppress the pronounced inelastic scattering background at small scattering angles,’ and all the presented results are energy-filtered. To underscore the importance of this filtering, the authors should compare 3D ED measurements obtained with and without the 10 eV energy-selecting slit.”*

Response: We agree that demonstrating the effect of elastic energy filtering on 3D ED data quality would be valuable for the reader. In response to the reviewer's suggestion, we have performed additional analyses comparing ED data (for two exemplary sample tilts) with and without the 10 eV energy filter, and we have included these results in the revised Supplementary Information. Specifically, we added a new panel in the SI (now Figure S10) that shows a direct comparison of diffraction patterns (and their intensity profiles) collected with vs. without the 10 eV slit. This comparison clearly illustrates that applying the energy filter dramatically reduces the diffuse background from inelastic scattering at low scattering vectors, thereby improving the signal-to-background ratio for weak diffraction peaks (particularly the small-angle peaks from lamellar stacking).

In the main text, we have added a sentence to alert readers to this comparison in the SI. In the Materials and Methods section (originally page 17, now page 24), where we originally stated “Zero-loss energy filtering is necessary to suppress the inelastic scattering background...”, we now follow up by referencing the new SI figure that demonstrates this point experimentally. We believe this addresses the reviewer's request: it provides concrete evidence of the benefit of the 10 eV energy-selecting slit.

Changes in Manuscript (Comment 3): We have updated the text in the Methods and the SI as follows:

- Original text (Methods, page 17, lines 400–404): “Zero-loss energy filtering is necessary to suppress the inelastic scattering background at small angles, which are crucial for investigation of molecular crystallites of typically nm-sized unit cells (thus small diffraction angles). For detailed information on the influence of EF and of the contributing volume, which changes during tilting, on the detected diffracted intensity see Figure S9.” (Note: In the original submission, Figure S9 referenced a different analysis; we have repurposed this number for the new filter comparison figure.)
- Revised text (Methods, page 24, lines 549-555): *“Zero-loss energy filtering is essential to suppress the strong inelastic scattering background at small scattering angles (which is crucial for detecting diffraction from molecular packing in ~nm-scale distances). We have quantified the impact of the energy filter by comparing ED data collected with and without the 10 eV slit: as shown in Figure S10, filtering improves the signal-to-background and signal-to-noise ratio of low-q diffraction features by effectively removing the diffuse background. Furthermore, we account for the changing probed sample volume upon tilting when processing the intensities (see Figure S11 for details).”* (Revised to include the new comparison)
- In the **Supplementary Information**, we added Figure S10 with a corresponding caption: *“Figure S10. Effect of energy filtering on ED data. Diffraction patterns acquired a*

without a 10 eV energy-filter slit (intensity scaled for better visibility and comparability) and b with energy filtering from the same DRCN5T:PC₇₁BM film at 0° and 75° tilt, and c line profiles comparing the two conditions. The unfiltered data show a significantly higher background at low scattering vectors (shaded region) due to inelastic electrons, which obscures weaker diffraction peaks. With elastic filtering, the background is markedly reduced, allowing clear detection of the (100) lamellar and (010) π - π stacking reflections (markings). This highlights the importance of energy filtering for accurate 3D ED of organic thin films.” (This figure and caption are new in the SI.)

These additions directly address Reviewer #1’s request by providing a side-by-side comparison of filtered vs. unfiltered diffraction data, thereby underscoring why we employed the energy filter in all measurements.

4. Comment (Reviewer #1): *“The reported 130 pA beam current for EELS acquisition seems high for such beam-sensitive materials. What were the electron dose rate and total dose during EELS measurements? Moreover, although the S L-edge map in Figure S8b exhibits a satisfactory signal-to-noise ratio, comparable to that of the C K-edge map, the S L-edge is barely visible in Figure S8c. It would be helpful for the authors to explain how they produced the S L-edge map despite its apparently low signal, particularly given the strong C K-edge. Additionally, the oxygen K-edge and nitrogen K-edge features are undetectable in the spectrum, raising further questions about measurement conditions and processing.”*

Response: We appreciate the reviewer’s close examination of our STEM-EELS data (Figure S8 in the original SI) and we address each question in turn:

- **EELS dose rate and total dose:** The 130 pA probe current for STEM-EELS was indeed chosen to obtain sufficient elemental signals and we acknowledge it is very high for beam-sensitive organics. We typically setup scan step 2 – 5 nm (depending on the feature size), which yields a dose rate of $3.2 \times 10^9 - 2 \times 10^{10} \text{ e}^-/\text{\AA}^2 \cdot \text{s}$, typical scan with 150 – 200 pixel squared grid and acquire EELS signals at pixel dwell time of 2 – 5 ms, which corresponds to an instantaneous electron dose between ~ 6500 and $101400 \text{ e}^-/\text{\AA}^2$ at each probed position, or between 46 and $28200 \text{ e}^-/\text{\AA}^2$ per scanned area. We have systematically studied the structure damage and reported that the key π - π stacking reflection (associated with the [010] direction) disappears after a cumulative dose of $\sim 5 \text{ e}^-/\text{\AA}^2$, whereas the lamellar (100) reflection remains detectable up to $>30 \text{ e}^-/\text{\AA}^2$ {Harreiß, C. et al., Solar RRL 2022, 6, 2200127; Wu, M. et al., Nature Communications 2022, 13, 2911}. We have added this finding to the Discussion (page 18) to clarify the DRCN5T:PC₇₁BM sample’s dose tolerance (and dose tolerance for the added second material system P3HT:PC₇₁BM). In practice, the STEM-EELS map was acquired with a dose far exceeding the threshold for preserving long-range structural order, meaning that the crystalline order was already destroyed during EELS acquisition. Nevertheless, the spatial distribution of the elements, particularly of heavier atoms such as sulfur, remains preserved as long as the atoms are not physically removed from the sample (i.e., no mass loss below the evaporation threshold) or subject to significant diffusion, which, based on our experience, is a valid assumption for OSC studies. Thus, reliable chemical mapping could still be obtained even though the crystalline order was lost.
- **Visibility of the S L-edge and how the S map was obtained:** It is correct that in the averaged EELS spectrum of the entire area (Figure S8c of the original submission), the sulfur L-edge signal appears very weak compared to the carbon K-edge. Nevertheless,

through careful post-processing we were able to extract the S signal for mapping, in particular, by employing the model-based fitting method developed by Verbeeck and van Aert {Verbeeck, J., Wan Aert, S., Ultramicroscopy 2004, 101, 207-224}, which is implemented in GMS 3. In the revised Manuscript, we have added a new Figure 6, highlighting the correlative aspect of TEM in a dedicated subsection of the results (see comment 2 above and also Comment 1, Reviewer #2). Here single pixel spectra extracted from two different sample regions, illustrate the fitting approach and the local presence of well distinguishable sulfur signal. In the supplementary information we have additionally included a note explaining that the S L-edge map was generated by fitting the signal from 110–410 eV (including background and both S-L and C-K edges) using a model-based fitting algorithm which is very robust against noise. This procedure yields the S distribution with acceptable clarity, as shown in Figure 6. We also mention that the S map's quality benefits from the fact that the S edge, though weak, is still well above the noise level thanks to the relatively high dose we used.

- **Absence of O K-edge and N K-edge in the spectrum:** We have added an explanation for this in the SI text. The O and N edges (at ~532 eV and ~400 eV, respectively) were not observed in our EELS spectrum for two main reasons: (1) Instrument and detector limitations: our EELS was performed using a Gatan UltraScan CCD camera, which has much lower detective quantum efficiency, as compared to state-of-the-art direct detection electron detectors, and also a higher noise floor, making the detection of O and N edges (which are intrinsically weaker in these organic samples) very challenging. (2) Acquisition settings were optimized for C and S – we chose a spectrometer dispersion and exposure that were ideal for the C K-edge (~285 eV) and S L-edge (~165 eV) region, in order to resolve and map those edges with good SNR. This meant that the higher-energy edges (O, N) did not receive the same signal amplification. In fact, we reference a prior study (cited in the revised SI, reference added) in which using a direct electron detector (Gatan K2) on a OSC sample system did reveal the N signal. With a more sensitive detector, nitrogen (and possibly oxygen) might be detected, but with our setup they were below the noise threshold.

All these clarifications have been added either to the main text or the Supplementary Information. In the Materials and Methods (page 24), we now state the electron fluence and area for each ED pattern and note that the critical dose ($\sim 5 \text{ e}^-/\text{\AA}^2$ for π - π stacking loss) was respected. In the Discussion section (page 18), we explicitly mention the dose at which crystallinity degrades, to reassure readers that our methodology took beam damage into account. In the new Figure 6 additionally single pixel EELS spectra illustrate the underlying raw data and fitting approach used for obtaining the shown elemental maps. And in the SI we have an expanded discussion of the EELS mapping conditions, including how the S map was derived and why O/N were not seen.

Changes in Manuscript (Comment 4):

- **Main manuscript:** On page 18, we inserted a sentence quantifying the damage thresholds: Original text (page 14, lines 302-304): “In this work, we demonstrated that beam damage can be effectively mitigated in 3D ED by... improved SNR.” Revised text (page 18, lines 376–383): “*In this work, we demonstrated that beam damage can be effectively mitigated in 3D ED by combining energy filtering with a targeted workflow (fresh regions for each pattern), resulting in significantly improved SNR. We also examined the dose limits for our material: notably, for DRCN5T the π - π stacking (010) reflection vanishes after a cumulative electron dose of $\sim 5 \text{ e}^-/\text{\AA}^2$, whereas the lamellar (100) reflection persists up to doses exceeding $30 \text{ e}^-/\text{\AA}^2$. For P3HT, both π - π and*

lamellar stacking reflections, vanish at a critical dose between ~ 5 and $15 \text{ e}/\text{\AA}^2$. We therefore designed our ED data collection to remain well below the $\sim 5 \text{ e}/\text{\AA}^2$ threshold per area, thereby ensuring the fidelity of the structural information.”

- **Main manuscript:** On page 16, a new Figure 6 was added, including STEM-EELS elemental maps and single pixel spectra showing the local presence of well distinguishable sulfur signal, described as follows in the corresponding figure caption: “Figure 6. Correlative TEM data. a The STEM-EELS S-L map shows the leaf-like structure of the DRCN5T which contains more sulfur, whereas the C-K signal represents the carbon-richer PC₇₁BM. Exemplary single pixel spectra from different locations illustrate the observable signal and the applied model-based fitting approach.... (continues describing remaining Figure content)”
- **Methods:** On page 24, we now explicitly give the ED and EELS acquisition parameters and dose: Original text (page 17, lines 404–410): “Due to beam sensitivity of the sample each diffraction pattern was acquired at fresh sample areas, using a fluence of $0.45 \text{ e}/\text{\AA}^2\cdot\text{s}$ and 1 s acquisition time, which ensures to stay below the reported critical dose. The illuminated area was $6 \mu\text{m}$ in diameter, of which an area with diameter of $\sim 3.4 \mu\text{m}$ contributed to the ED pattern due to the SAD aperture size. The sample translation step size between ED pattern acquisitions was $10 \mu\text{m}$. STEM-EELS data were acquired for investigation of the nanomorphology, a scanning grid of 144×144 with a probe step size of $\sim 3 \text{ nm}$, a dwell time of 0.008 s and a beam current of 130 pA was used.” Revised text (page 24, lines 556–566): “...each diffraction pattern was acquired from a fresh area (illumination diameter $\sim 6 \mu\text{m}$, contributing area $\sim 3.4 \mu\text{m}$ via SAD aperture) using a beam flux of $\sim 0.45 \text{ e}/\text{\AA}^2\cdot\text{s}$ for 1 s. This corresponds to $\sim 0.45 \text{ e}/\text{\AA}^2$ per pattern, well below the $\sim 5 \text{ e}/\text{\AA}^2$ critical dose for maintaining π - π crystallinity. The sample was translated $\sim 10 \mu\text{m}$ between acquisitions to avoid cumulative damage, resulting in $\sim 5000 \mu\text{m}^2$ sampled area for tilt series acquisition. The STEM-EELS elemental mapping was performed after completing the ED tilt series: a 100 pA probe with 5 ms dwell and 4.5 nm step yielded a dose rate of $\sim 3.08 \times 10^9 \text{ e}/\text{\AA}^2\cdot\text{s}$, summing to $15407 \text{ e}/\text{\AA}^2$ instantaneously (i.e. per scan point) or $94 \text{ e}/\text{\AA}^2$ over the full 200×200 scan frame, meaning that long-range crystalline order was already destroyed during EELS. However elemental composition (particularly for heavier elements like S) remains as long as the elements are not physically removed from the sample (no mass loss below the evaporation threshold), or diffused.”
- **Supplementary Information:** We expanded the discussion on STEM-EELS. Original SI text: did not explain the S, O, N edge issues in detail. Revised SI additions: “...The S L-edge map was extracted by fitting the signal from ~ 110 – 410 eV using the model-based fitting algorithm as implemented in GMS², which yields a discernible map due to the optimized acquisition conditions used. Note: O K-edge ($\sim 532 \text{ eV}$) and N K-edge ($\sim 400 \text{ eV}$) signals were not detected in our EELS dataset because the acquisition parameters were tuned for the C-K and S-L edges and the limited CCD detector’s efficiency and high noise. In a separate test on a similar OSC sample, a direct electron detector was able to capture the N K-edge under comparable conditions.³”

Through these clarifications and additions, we have addressed the reviewer’s concerns: we specify the electron dose used for EELS, justify how the sulfur map was obtained despite a weak signal, and explain why oxygen and nitrogen edges were not seen. We trust that this additional information will assure readers that our methodology was carefully optimized for these beam-sensitive samples and that the data interpretation is sound.

Reviewer #2 (Comments and Responses)

We thank Reviewer #2 for the critical assessment of our manuscript. We appreciate the frank feedback on the novelty and scope of our work. Below, we provide a detailed response to all points raised. We believe that the changes made (including additional experiments and a clearer discussion of the advantages and limitations of 3D ED vs. GIWAXS) have strengthened the manuscript significantly.

1. Comment (Reviewer #2): *“The authors highlight the primary advantage of 3D ED as enabling the combination of molecular packing and morphological characteristics within the BHJ film. Nevertheless, the manuscript did not clearly discuss the specific correlation between these morphological features and molecular packing, nor explain why this point is significant.”*

Response: We thank the reviewer for pointing out the need to more explicitly discuss the correlation between morphology and molecular packing in our 3D ED/TEM analysis, and its significance. In the original submission, we had already observed certain correlations—for example, we noted that the DRCN5T donor domains exhibit a “leaf-like” morphology (visible in TEM images and evident in STEM-EELS elemental maps, shown as insets in Figure 1) and many domains contain multiple crystalline grains as revealed by 4D-STEM. However, we realize that we had not clearly articulated the implications of these correlations in the text. To address this, we have expanded our discussion in the Results section to clearly connect morphological features to the molecular packing/orientation findings, and we now explain why these correlations are scientifically important.

Concretely, in the revised manuscript we added a dedicated subsection in the Results, including a new figure (page 15–16), that explicitly examines what can be learned by correlating real-space morphology with diffraction (reciprocal-space) information in these OSC films. We now discuss, for example, that the elongated (leaf-shaped) donor-rich domains in our system tend to exhibit edge-on molecular orientation, with the π - π stacking plane normal along their short axis (as evidenced by our 4D-STEM orientation mapping), whereas more isotropic (rounded) domains correlate with face-on orientations. We explain that such correlations are significant because the relative orientation of crystalline domains with respect to the phase morphology can strongly influence charge transport pathways in the solar cell. For instance, if π - π stacking aligns along certain directions of the domain shape, it could facilitate charge transport along those directions. We also note that linking morphology and molecular packing provides insight into the effects of processing (such as solvent vapor annealing) on the film structure-property relationships.

Furthermore, we have ensured that specific data from our measurements are cited to support these correlations. We refer to Figure 1 and the new Figure 6, where the morphology is characterized, and to Table 1 (and the SI), where orientation and packing are quantified. In this way, we emphasize that our correlative approach provides multi-faceted insight—for example, domain size from TEM images, crystallite size from diffraction (ED, 3D-ED), orientation relationships from 4D-STEM – all on the same identical sample.

In summary, the revised text now clearly articulates the correlation between nanomorphology and molecular packing in the BHJ film and why this is important. This directly addresses the reviewer’s comment by making an implicit point explicit: the true power of 3D ED in the TEM lies not just in obtaining “X-ray-like” structural data, but in doing so at specific locations that can be directly tied to the morphological context. This capability enables a deeper

understanding of how morphology and crystallography together influence device-relevant properties.

Changes in Manuscript (Comment 1 – Reviewer #2): We have added a new subsection on page 15 in the Results and adjusted wording elsewhere. For example:

- Original text (page 9, lines ~270–278): “Comparing these sizes with the CCLs indicates that each domain comprises multiple crystallites, separated by grain boundaries. Our prior diffraction imaging studies... confirmed the leaf-like domain morphology (Figure S8) with DRCN5T domains measuring ~79 nm (long axis)... etc. Each domain comprises multiple crystallites...” (The original text described the observation but did not explain significance).
- Revised text (pages 15-16, lines 294-325): “*A major strength of 3D ED is its integration in modern TEM platforms, which facilitates seamless transitions between diffraction, imaging, and spectroscopy for comprehensive and correlative characterization (cf. Figure 1). Notably, this correlative real-space/diffraction approach allows us to link morphological features with molecular packing orientation. Figure 6 shows a correlative dataset acquired from a single sample in TEM, comprising STEM-EELS, 4D-SCED and 3D ED. STEM-EELS provides elemental mapping and reveals donor-acceptor phase separation through their chemical composition: the small molecule contains more sulfur, while the fullerene acceptor is richer in carbon. The corresponding S-L and C-K maps obtained via model-based fitting⁴⁸, resolve leaf-shaped donor domains (cf. Figure 6a, see SI for details on evaluation). Additionally, our 4D-SCED analysis locally reveals the presence and relative orientations of edge-on and face-on domains through the in-plane π - π stacking and (100) peaks (cf. Figure 6b). The π - π stacking orientation map shows that the leaf-shaped regions consist of edge-on oriented DRCN5T crystallites with their π - π stacking plane normal aligned along the domain’s short axis. In contrast, smaller or more rounded domains mainly host more face-on oriented crystallites.³⁹ This specific correlation between domain shape and molecular packing (edge-on vs. face-on) is significant; it indicates that solvent-vapor annealing induces alignment of the crystalline domains with the mesoscopic phase separation structure. Such alignment can influence charge transport pathways: edge-on oriented domains (π - π stacking direction in-plane) may facilitate in-plane transport across larger distances, whereas face-on domains promote vertical transport.⁴⁹ Furthermore, the PC₇₁BM also contributes to the 4D-SCED diffraction signal as amorphous halo, as already discussed for the SAED patterns. The corresponding signal is mapped by applying an annular virtual detector, revealing the acceptor distribution via underlying diffraction data, as an alternative way to reveal the acceptor distribution via analytical information in STEM-EELS. By combining 3D ED with complementary analytical and (diffraction-)imaging methods, we directly reveal how nanomorphology and molecular orientation coincide in the BHJ film, an insight that pure GIWAXS, lacking real space information, cannot provide. In our case, we find that each “leaf” domain (~80 nm long) contains multiple smaller crystallites (~20 nm, per Scherrer analysis, cf. Table 1), indicating a mosaic structure within domains. Additionally, 3D ED provides access to the degree of crystallite alignment, capturing inclined crystallites, that deviate from strict edge-on or face-on orientation. The dataset also reveals an isotropic fraction of crystallites, with deviations from strict orientation described as mosaicity (cf. Table 1 and SI). Pinpointing these relationships (domains vs. crystallites, orientation vs. shape) underscores the value of a correlative TEM approach for understanding structure–property relationships in OSCs.*”

This dedicated subchapter and new figure directly tackle the reviewer's comment by describing the correlation and its importance. We trust that this makes the novelty of our approach clearer: 3D ED is not meant to replace GIWAXS, but rather to offer deeper insight by correlating structure with local morphology (something that is indeed significant for complex functional materials like BHJ films).

2. Comment (Reviewer #2): *“While 3D ED has proven powerful for microcrystalline particles (MOFs, COFs, etc.), this reviewer believes it cannot replace GIWAXS for film structure analysis. This is due to several limitations on ED: 1) The testing area of ED is only nanometer-scale, whereas the GIWAXS beam spot can be tens of micrometers; 2) ED cannot probe directions close to the in-plane direction (GIWAXS uses incident angles $< 1^\circ$); 3) The requirements for in situ ED studies are more stringent (GIWAXS can be readily integrated with various stages, such as heating/cooling and optical microscopes). More importantly, X-ray scattering has long enabled clear diffraction signals in both horizontal and vertical directions from large-area organic thin films and has successfully resolved their 3D structures (Science 2010, 330, 808; Science 2015, 348, 1122). Therefore, while acknowledging this paper as the first trial of 3D ED on a BHJ film, the reviewer concludes that, compared to the well-established GIWAXS technique, it does not constitute a breakthrough or demonstrate convincing advantages.”*

Response: We appreciate that GIWAXS is a well-established and foundational technique for analyzing organic thin films, and we had already emphasized this point in our original submission. However, some of the technical assertions raised in this comment appear to reflect misunderstandings of transmission electron microscopy and diffraction and/or an incomplete consideration of the explanations provided in the manuscript. In particular, one point seems to arise from a misconception regarding the orientation of the Ewald sphere in 3D ED at 0° tilt angle (see point (2) below for details). Furthermore, contrary to what the reviewer's comment suggests, our study does not claim that 3D ED may 'replace' GIWAXS. Rather, it emphasizes the complementarity of the two methods and highlights capabilities of 3D ED that extend beyond what GIWAXS alone can provide. Specifically, because 3D ED is performed within a modern TEM, it allows reciprocal-space sampling, real-space imaging, and the acquisition of analytical signals (EELS/EDXS) for nanomorphology and compositional analysis, as well as diffraction imaging (4D-STEM / 4D-SCED) for correlated nanoscale real- and reciprocal-space characterization—all on the same microscopic region. This multimodal correlation represents the central advance of our study, and the new data and clarifications added in the revision (including an additional OSC system, see Response to Comment 3 below) substantiate its broader applicability. We now address each of the reviewer's specific points (1–3) regarding the limitations of ED:

- **(1) Probed area (nanometer vs micrometer scale):** While TEM is conventionally used to probe nanometer-scale regions, our use of selected-area or low-magnification parallel-beam TEM diffraction enables sampling over micron-scale areas, rather than being restricted to nanometer-scale as stated by the reviewer. In the present work, each tilt pattern integrates diffraction from a $\sim 3.4 \mu\text{m}$ diameter selected area ($6 \mu\text{m}$ illumination, with the SAD aperture defining the contributing region). Modern TEM optics can in fact provide much larger selected areas ($\geq 10\text{--}20 \mu\text{m}$ diameter) with quasi-parallel illumination if required. By switching to field-free mode (objective lens off, Lorentz lens on), illumination of sample areas $> 100 \mu\text{m}$ in diameter and collection of large angle diffraction signals have already been demonstrated (cf. reference {Kang,S.

et al., Nature Communications 2025, 16, 1305}), including for gold nanoparticles as shown in Figure S1 of that work. This directly refutes the notion that ED must be limited to nanometer-scale sampling. Importantly, for statistically representative texture information, 3D ED can average over thousands to millions of crystallites in a single pattern—comparable in representativity to many micro-beam GIWAXS experiments—while still allowing subsequent nanoscale correlative investigation (e.g., zooming into sub-domains by imaging or 4D-STEM). Moreover, by distributing the tilt series over ~ 150 fresh positions, we effectively sampled $\sim 5 \times 10^3 \mu\text{m}^2$ of film in one 3D dataset. This hybrid “distributed acquisition” strategy has no analogue in GIWAXS and preserves structural fidelity under stringent dose budgets.

- **(2) Access to near in-plane directions:** The reviewer suggests that ED cannot probe directions near the film plane ($q_z \approx 0$). We believe this reflects a misunderstanding of the diffraction geometry in TEM. Because of the extremely short electron wavelength and the transmission geometry of TEM, the ED pattern at 0° tilt directly provides in-plane information. In reciprocal-space terms, the effectively flat Ewald sphere (arising from its very large radius) perpendicular to the incident beam means that the entire diffraction pattern at 0° corresponds to scattering vectors lying in the plane of the film, i.e., in the q_x - q_y -plane at $q_z = 0$. By contrast, GIWAXS cannot access exactly in-plane scattering ($q_z = 0$) due to the substrate horizon and beam geometry: the incident angle can be made small, but not zero. To avoid misunderstanding, we have now explicitly countered the reviewer’s point (2) in the manuscript by rephrasing a sentence in the Discussion (page 18) clarifying that due to the transmission geometry, ED naturally captures in-plane (q_{x-y}) information at 0° tilt with no missing horizon, yielding better-defined in-plane peaks than GIWAXS in our experiments.
- **(3) In situ compatibility and environmental control:** Yes—GIWAXS currently provides simpler real-time environmental control (e.g., solvent vapor flow, thermal ramps, large-area averaging), and we explicitly acknowledge this limitation. However, 3D ED can already leverage rapid sparse-tilt strategies (e.g., $0^\circ + \text{high tilt}$) for quasi-dynamic snapshots, and emerging MEMS-based TEM holders (heating, bias, gas) maintain large tilt ranges, enabling time-resolved or temperature-dependent reciprocal-space sampling. Importantly, during such studies the same region can be re-imaged, chemically mapped, and locally re-diffracted, enabling spatiotemporally resolved correlative analysis that GIWAXS alone cannot provide.

At the end of this comment, the reviewer highlights the well-established success of GIWAXS and questions whether our work constitutes a ‘breakthrough’ or demonstrates convincing advantages over GIWAXS. We fully acknowledge these landmark studies and now cite one of them explicitly in the Introduction as benchmark. Our strategy deliberately anchors 3D ED metrics (lattice spacings, CCLs, mosaicity) against GIWAXS measurements obtained on the same films (both lab + synchrotron), thereby using GIWAXS as a calibration reference while revealing where 3D ED provides further insight—for example, resolving local crystalline grains within a single morphological “leaf” and establishing orientation–morphology correlations.

In the Introduction (pages 3-6) and Conclusion (page 22), we acknowledge that GIWAXS is a highly developed technique and that our experiments in fact use GIWAXS as a benchmark to validate 3D ED. We then emphasize that the added value of 3D ED lies in the integration of multiple techniques within a single instrument on the same sample—something that GIWAXS cannot replicate. While GIWAXS can certainly be combined with other methods, this is typically not achievable at the same spatial resolution and/or on the exact same microscopic region. Instead of stating this explicitly, we chose to phrase our conclusion in a deliberately

cautious manner: namely, that having both GIWAXS and 3D ED expands the toolkit for investigating organic solar cells, and that the “breakthrough” lies in integrating complementary methods to get a more complete picture.

To further convince the reviewer and readers of the value of 3D ED, we have taken additional steps in the revision (also related to Comment 3 below). Specifically, we added a second example OSC system, the extensively studied P3HT:PC₇₁BM blend, to demonstrate that the applicability of 3D ED is not limited to a single material and that it can reproduce well-known structural changes upon thermal annealing—a classic OSC processing-structure relationship extensively characterized by GIWAXS. This addition directly addresses the concern about broader applicability and illustrates that 3D ED provides scientifically significant insights, such as monitoring structural evolution resulting from post-processing.

3. Comment (Reviewer #2): *“Furthermore, the manuscript fails to demonstrate the broader applicability of 3D ED to different organic solar cell systems or its scientific significance through structure-property relationship studies.”*

Response: In order to address this point, we have conducted additional experiments on a second organic solar cell model system and we discuss a structure–processing–property relationship. Specifically, we chose the prototypical P3HT:PC₇₁BM bulk-heterojunction system for further demonstration. P3HT:PC₇₁BM is a well-studied OSC blend where the effects of thermal annealing on structure (morphology and molecular ordering) have been extensively characterized by GIWAXS and correlated with solar cell performance. This makes it an ideal test-case to show that 3D ED can be applied beyond our original system and can capture meaningful structural changes that relate to processing.

New experimental data (P3HT:PC₇₁BM): In the revised manuscript, we introduce new 3D ED results for a P3HT:PC₇₁BM film, both in as-cast state and after thermal annealing. We found that 3D ED can detect the known structural evolution: upon annealing, P3HT crystallites grow and the degree of (100) stacking order increases, similar to what GIWAXS reports in literature (we cite relevant studies). We include a comparison showing that our 3D ED measurements of lattice spacing and texture in P3HT:PC₇₁BM match literature values from GIWAXS, reinforcing that 3D ED is generalizable. More importantly, we use this example to illustrate a structure-property connection: thermal annealing improves P3HT crystallinity and phase separation, which is known to enhance charge transport and device efficiency – our 3D ED data directly captures the structural side of this relationship.

In the Discussion/Outlook, we also explicitly draw attention to how this second example serves as a structure–property case study. We state that by using 3D ED on P3HT:PC₇₁BM, we essentially reproduce what was known from GIWAXS (e.g., improved (100) lamellar stacking upon annealing correlates with improved charge transport in those layers), thereby validating 3D ED in a different context and showing it can be used to probe processing-induced structural changes – a key aspect of structure-property studies.

Additionally, beyond adding a new material example, we have broadened our discussion in general terms: we mention in the Conclusion that 3D ED can be applied to “a multitude of other nanostructured materials” (a claim from our abstract that we now back up with the P3HT:PC₇₁BM data and the reasoning given in response to Reviewer #1 about perovskites and

others). We also highlight that our methodology could be used to study any polycrystalline thin film where local structure is of interest, not just organic solar cells.

In summary, we have directly addressed the reviewer's concern by demonstrating another OSC system and by explicitly discussing a structure-processing relationship (thermal annealing) in that system. We believe this significantly strengthens the paper, as it not only appeases the reviewer but also makes our work more broadly interesting to the community.

Changes in Manuscript (Comment 3 – Reviewer #2):

- In the **Results**, near the end (prior to Discussion), we added a subsection introducing the P3HT:PC₇₁BM findings: Revised text (pages 16-17, lines 338–350): *“To demonstrate the broader applicability of 3D ED, we have additionally applied our method to a second OSC system – the archetypal P3HT:PC₇₁BM blend – and observed the expected structural changes upon thermal annealing (cf. Figure 7). In summary, 3D ED analysis of P3HT:PC₇₁BM films before and after annealing reveals an increase in P3HT (100) lamellar stacking ordering, evidenced by the more clearly defined (100) ring and the appearance of higher order rings (see 0° rotational average and sphere average in SI). Moreover, even the untreated sample shows texture, consisting of edge-on, face-on and isotropic oriented crystallites, as indicated by the azimuthal intensity of the (100) and π - π stacking rings (see azimuthal χ -cuts in SI). The thermally annealed sample exhibits a comparable texture, but shows sharper and more intense peaks, indicative of larger crystallite sizes, and higher crystallinity. Those findings are consistent with the enhanced crystallinity and phase segregation known to improve device performance in this system.⁵⁰⁻⁵² The fact that 3D ED can quantitatively capture these changes (in agreement with prior GIWAXS studies on the same system) confirms that our approach is general and can be used to study structure–property relationships in organic photovoltaic materials.”*
- In the **Conclusion/Outlook** (page 22), we incorporate the broader perspective: *“The demonstration of our method on a second OSC system (P3HT:PC₇₁BM) resolving structural change upon post-processing exemplifies its potential, and we foresee applying 3D ED to other organic, hybrid, and inorganic polycrystalline thin films where its unique combination of local and reciprocal-space analysis can yield fresh insights.”* (Conclusion, page 22, lines 498-501; new content emphasizing broader use.)

4. Comment (Reviewer #2): *“Finally, the first eight pages devote excessive space to background information together with Figures 1 and 2, while the experimental data in the manuscript (as well as supporting data in Figs. S1–S10) appear too weak and thus fall short of the standard for a research paper.”* (Reviewer #2's summary decision was: *“In light of the above comments, the paper would not be recommended for publication in Nature Communications.”*)

Response: We respectfully disagree with the reviewer's statement regarding the 'excessive space' devoted to 'background information'. Nature Communications has a broad readership, and we expect our paper to attract readers from diverse fields—including the broad OSC community, for which 3D ED is a completely new methodology, as well as the more specialized (but also large) electron microscopy and X-ray scattering communities. A clear description of the basic principles of 3D ED applied to polycrystalline films, its scattering geometry, and data evaluation is essential to convey the essence of the methodology and to facilitate its broader

acceptance and application. Moreover, as the reviewer's comments indicate some confusion regarding how in-plane information is obtained from 3D ED, this further underscores the importance of providing a thorough methodological description. Finally, we note that neither of the other two reviewers raised concerns about the length of the background; indeed, Reviewer #3 explicitly stated that **“the manuscript is well written and clearly presented”** and that it provides an **“appropriate level of detail for others to adopt this approach”**.

At the same time, we agree with the reviewer that additional experimental data are important to demonstrate the broader applicability of the method and to highlight the direct correlation of 3D ED with other TEM techniques. In the revision, we have made a concerted effort both to extend and strengthen the presentation of our experimental results and to streamline the introductory/background sections. The key changes we implemented are as follows:

- **Enriching and better integrating Figure 1 with results:** As described in response to Reviewer #1 Comment 2, we have reworked Figure 1 to emphasize that it presents real experimental data acquired from the same sample studied with 3D ED, rather than merely example images illustrating the different methods (4D-SCED, EF-SAED, STEM, STEM-EELS).
- **Adding new experimental content (as detailed above):** The inclusion of the P3HT:PC₇₁BM additional data, along with the new comparisons (e.g., filtered vs unfiltered ED, see Comment 3 Reviewer #1), strengthens the paper by contributing more substantive results. We have also carefully verified that all supporting figures (S1–S11) are now clearly described and contextualized in the text so that they directly support the paper's claims. For example, we reference Figure S10 for filtering (as noted above), Figure S9 for sample flatness, and Figure S5 for GIWAXS vs ED profiles, ensuring that each supplementary figure plays a clear and specific role in the overall argument.
- **Streamlining background text:** We have carefully revised the Introduction, trimming redundancies to make it more concise. Together with the modifications to Figure 1 described above, the narrative now transitions into the results more naturally.

We hope that the numerous additions and improvements we have made will convince Reviewer #2 that the manuscript now meets the high standard expected for publication in Nature Communications. In the revision, we have not only clarified and defended the novelty of our approach but also provided additional proof of its validity and utility. The dataset is now richer—covering two material systems and multiple modes of analysis—and more clearly connected to key questions in the field, such as processing effects and multi-modal correlation.

Reviewer #3 (Comments and Responses)

We thank Reviewer #3 for the very positive feedback and the minor revision suggestions. We are pleased that the reviewer found our results impressive and the comparisons clear. We have addressed the two main points raised, as well as the comment on code availability, as detailed below.

1. Comment (Reviewer #3): *“From reading the manuscript, the main advantage of the 3D ED approach (especially compared to GIWAXS) seems to be the ability to combine it with other TEM-based characterization, such as imaging or EELS. Given the need to translate the sample between ED images to avoid beam damage, the finite size of the electron spot, and the changing footprint during rotation, additional discussion of the feasibility of spatially resolved 3D ED*

would be welcome. Does this present a fundamental limit to the size of different regions which can be measured in a film, and how does this compare to the features which might be of interest especially for OSCs?”

Response: This is an insightful question. The reviewer emphasizes that one key advantage of our approach is the potential for spatially resolved structural analysis (probing different regions in a film). We agree that it is important to discuss the practical limits of “mapping” a film with 3D ED, especially in comparison to GIWAXS.

The feasibility of spatially resolved 3D ED is indeed constrained by two main factors: beam damage (dose) and sample homogeneity requirements, as the reviewer alludes. We have now expanded our Discussion (page 20) to address this point explicitly:

- We explain that **for highly beam-sensitive materials (like OSC blends), one cannot collect a full 3D ED tilt series on a single small region** because the cumulative dose would destroy the order in that region. That is why we spread the tilt series over many positions (as already described in the original manuscript). This inherently trades spatial resolution for completeness of reciprocal space. If one wanted to examine a specific microscopic region of interest, one could only do a limited number of tilts on it before damage sets in. We reference our earlier work {Fürk, P. et al., Journal of Materials Chemistry 2023, 11, 8393-8404} where we demonstrated a “sparse tilt” approach: taking just two tilt angles (0° and $\sim 75^\circ$) to rapidly feedback the in-plane and out-of-plane π - π stacking character of the film. This approach sacrifices full 3D information but yields partial structural information (in-plane vs. out-of-plane), thus destroying only small region, promising for achieving spatial mapping. We now discuss this as a viable strategy for spatially resolved 3D ED in page 20 (and we had indeed touched on it in the Outlook originally, now we highlight it as an answer to the reviewer’s query).
- We compare this to GIWAXS: GIWAXS inherently averages over a mm-scale area, or uses micro-beam GIWAXS to get tens to hundreds of μm resolution, so it’s not spatially resolved at the nanoscale. We mention that spatial mapping of nanostructures at the ~ 100 nm scale is a strength of TEM, whereas GIWAXS can map on the micron scale with specialized setups. For features of interest in OSCs (phase separation domains ~ 10 – 100 nm), 3D ED combined with imaging (4D-STEM) can in principle target individual domains, which GIWAXS cannot differentiate – however, doing so in practice requires careful dose management as noted.
- We reaffirm that currently, our full 3D ED was done on an ensemble of areas (not one contiguous area), but if one were interested in spatial heterogeneity, one could do fewer tilts on a grid of positions to probe differences. We now explicitly state this in the text.

In short, the revised manuscript (page 20) contains a discussion on spatial resolution: acknowledging the dose-limited trade-off and referencing the idea of sparse tilt acquisitions for mapping structural inhomogeneities. We believe this directly addresses the reviewer’s question.

Changes in Manuscript (Comment 1 – Reviewer #3): On page 20, we added:

- Original text: (The original Outlook briefly mentioned something similar in general terms, but it wasn’t prominent in results/discussion.)
- Revised text (page 20, lines 432–453), addressing feasibility of spatially-resolved 3D ED: “(discussion on in situ capabilities of 3D ED) ... *The possibility of capturing sparse snapshots of 3D reciprocal space is not only a valuable approach for achieving higher*

temporal resolution in in situ studies, but also for increasing spatial resolution of our 3D ED approach: In our demonstration, the 3D ED data represent a spatially averaged structure (covering $\sim 5000 \mu\text{m}^2$), as the tilt series were distributed across the sample to minimize beam damage at any single location. Since homogeneity was experimentally confirmed in our OSC sample systems, this averaging approach is justified. For application in OPV devices homogeneity is generally required. Importantly, 3D ED is in principle sensitive to local inhomogeneities and could therefore serve as a valuable tool to identify and evaluate them. However, achieving spatial mapping of structural and crystallographic texture variations by acquiring complete 3D ED tilt series on a sample raster is extremely time-consuming and challenging, as the electron dose would quickly reach prohibitive levels, depending on the targeted spatial resolution. By limiting the tilt series to only the most informative angles, both measurement time and beam damage can be drastically reduced, while spatial resolution can in turn be enhanced. For example, our previous study demonstrated that collecting ED patterns at only two tilt angles (0° for in-plane and $\sim 75^\circ$ for a near out-of-plane view) provided the most essential structure information, which could be in principle spatially resolved if probed at different locations across the OSC film. Such a sparse sampling strategy enables mapping of regions of interest – separated by micrometers – while keeping the electron dose per region low. In essence, 3D ED can be tailored to spatial resolution by trading off angular resolution – an approach particularly useful for investigating inhomogeneous samples. This compares favorably with GIWAXS, which typically averages over much larger areas (millimeter-scale), unless synchrotron micro-beam techniques are employed. Thus, while full 3D reciprocal-space mapping is dose-limited to an averaged region, targeted partial 3D ED scans can be performed on different locations to reveal spatial heterogeneities relevant to OSCs.”

2. Comment (Reviewer #3): “The film was removed from the substrate for 3D ED measurements; with this in mind, a more detailed discussion of the sample limitations for this technique seems appropriate. This is briefly touched on in the discussion section.”

Response: We agree that we should be more explicit about the sample preparation requirements and current limitations of 3D ED. In the original manuscript, we noted in the Methods the need to detach the film and briefly mentioned in the Discussion/Outlook that obtaining out-of-plane info is hard without cross-section or tilting (hence our use of freestanding, tilted film). However, we did not explicitly enumerate the associated limitations.

In the revision, we have added a short, dedicated paragraph on “Sample prerequisites” in the Discussion/Outlook (on page 21, just before the Conclusion), which covers:

- **Need for substrate removal or TEM-compatible substrate:** We note that because our method requires tilting the sample in free space, it must be removed from its original substrate. While we successfully achieved this for a glass substrate using a water float-off, this approach may not be possible for all substrate types or all materials. Alternatively, 3D ED can be applied to films that are grown directly on TEM-compatible, electron transparent membranes.
- **Sample flatness and integrity:** Because the sample must be tilted to approximately $\pm 75^\circ$, the film needs to remain flat and avoid curling or warping once removed from the substrate and mounted on supports. Our floating technique (detaching via dissolving an sacrificing layer) worked well for this material, but other systems may be more fragile

or difficult to detach. If a film after transfer is not perfectly flat, the effective incidence angle is determined not only by the holder tilt but also varies across the sample. When fresh sample regions are used for each tilt (due to dose limitations), such variation can compromise the 3D reconstruction. Thus, even if a film can be transferred onto a TEM support, there remain potential limitations for applying 3D ED.

- **Electron transparency (thickness):** The film or sample must be electron-transparent (typically $< \sim 150$ nm thick for 300 kV electrons) to collect ED patterns. This implies that very thick films or multi-layer stacks may not be suitable unless thinned, e.g., by ultra-microtomy or focused ion beam, though such approaches introduce other issues such as damage. In addition, one must consider that the effective sample thickness encountered by the electron beam increases with tilt angle. We now explicitly state this as a limitation.
- **Field of view vs. representativity:** We acknowledge that because our method probes a relatively small area (even if micron-scale, still smaller than device scale), it is important to assure that the sampled region is representative or to examine multiple regions (which we have done). This is more of a caution than a strict limitation, but we consider it worth mentioning.

We believe that spelling this out helps readers and addresses the reviewer's point by clarifying practical considerations: 3D ED in TEM is not as straightforward as GIWAXS in terms of sample preparation, and this should be acknowledged.

Changes in Manuscript (Comment 2 – Reviewer #3): On pages 21-22 (just before Conclusion), we added:

- Revised text (pages 21-22, lines 468–485), addressing sample preparation prerequisites: “(discussion on applicability to other beam-sensitive materials) ... *Beyond these material inherent characteristics, sample preparation represents another key factor for extending the applicability of our 3D ED approach to other thin films. A practical prerequisite of the method is the use of electron-transparent, free-standing samples. In our study, this was achieved by detaching the OSC films from their substrate and transfer onto a TEM grid. This step, however, may not be feasible for all material systems: fragile films are prone to cracking or warping during removal, while some may lack a convenient sacrificial layer (such as PEDOT:PSS) to facilitate release. In addition, the film must remain sufficiently flat on the TEM grid, as any curvature can distort the tilt geometry and complicate data interpretation (cf. Figure S9). Another prerequisite concerns sample thickness: the technique requires electron transparent regions typically in the range of ~ 100 nm, depending on the material. Moreover, it is important to account for the increase in projected thickness during tilting. For thicker films or multilayer, selective plan-view thinning or cross-sectioning is required. In this context, combining 3D ED with double-wedge preparation is highly promising, as it enables depth-dependent analysis of structural and textural evolution across thicker films.⁶³ While originally developed for inorganic systems, this approach may also be extended to organic films. For cross-sectioning, focused ion beam (FIB) lift-out or microtomy can be employed. Care must be taken, however, to minimize process induced damage that could alter the native structure. We note these requirements to emphasize that, in its current implementation, our 3D ED approach is best suited for films that can either be fabricated directly on TEM-compatible membranes or transferred to them without significant perturbation.*”

Comment on Code Availability (Reviewer #3): “While I have briefly looked through the code I am not expert enough to evaluate it. However, a README with instructions for how to get started with using this should be included.”

Response: We appreciate the reviewer’s effort in examining our provided code and the suggestion to improve its usability. In response, we have added a README file with usage instructions to our code repository (linked in the Supplementary Information). The README includes guidance on installing necessary packages, loading the example datasets we provided, and running the scripts to reproduce the key analyses (e.g., processing a tilt series into a 3D reciprocal-space map, extracting q_x – q_z maps, etc.). We agree that this addition will make it easier for others—even those not intimately familiar with diffraction data processing—to get started with our code and apply it to their own data.

In the revised **Supplementary Information**, under the Code Availability section, we now explicitly state that a README and step-by-step instructions have been included, as recommended by the reviewer. This ensures that any reader accessing the code will be aware that detailed documentation is available.

Changes in Manuscript (Code Availability): In the **Code Availability** statement (at the end of the manuscript/SI):

- Original text: “The code used for 3D ED data processing is available in SI.”
- Revised text: “*The code used for 3D ED data processing is available in SI. We have added a README file with step-by-step instructions to facilitate use of the code by the community.*”

In conclusion, we once again thank all three reviewers for their valuable feedback. We have made extensive revisions to the manuscript and supplementary information to address every point raised: clarifying our arguments, adding new data, and improving the overall quality of the paper. We believe the manuscript is much improved as a result of this revision process.

We hope that the reviewers will find our responses satisfactory and the changes adequate, and we kindly request a favorable reconsideration of our manuscript for publication in **Nature Communications**.

Response to Reviewer Comments

Dear Editor and Reviewers,

We sincerely appreciate the time and effort from Reviewers and Editor invested in evaluating our work. We are encouraged that Reviewers #1 and #3 found our revised manuscript greatly improved and recommended for publication. We thank Reviewer #2 for acknowledging our adjustments based on the previous revision. However, we respectfully, yet firmly disagree with the final judgement of Reviewer #2 and remain convinced that our revised manuscript represents a significant piece of research, not least due to the positive response of the other Reviewers. Furthermore, we want to thank Reviewer #1 for providing additional comments and insights with respect to comments of Reviewer #2. In hindsight of the additional constructive feedback of Reviewer #1, we aim to resolve any remaining reservations of Reviewer #2 concerning the novelty and structure of our work. We carefully revised the manuscript and supplementary information by modification of figures and text, and included additional data. In the following point-by-point response we hope to address all concerns, by including clarifications of misconceptions and indicating our careful additional revisions.

Due to the close relationship of the comments of the respective reviewers, one being a perspective on the other one, we structured our response as follows: The comments of Reviewer #2 precede the corresponding remark of Reviewer #1, both *italicized* for clarity. Our response follows, including indications of revisions in the manuscript where necessary.

1.Comment Reviewer #2: *“The 3D ED method is not a novel characterization technique for determining crystal structures of nanomaterials where single crystals are unavailable. Its application in the OSC field. For instance, the identification of molecular packing in P3HT by 3D ED has been well established for over a decade, as evidenced by the pioneer works of Martin Brinkmann and co-workers (e.g., *Macromolecules* 2007, 40, 7532; *Macromolecules* 2010, 43, 4961; *Adv. Funct. Mater.* 2016, 26, 408; *Mater. Chem. Front.* 2020, 4, 1916). In light of their studies, the generality of 3D ED that the authors aim to demonstrate in this work appears less meaningful.”*

Remark Reviewer #1: *“Regarding comment 1, as a researcher specializing in transmission electron microscopy, I find the 3D ED approach demonstrated by the authors to be impressive. Its most compelling aspect is its compatibility with other advanced S/TEM techniques, including 4D-STEM, in situ TEM, and ptychography, enabling a comprehensive suite of tools for materials and device studies. The inclusion of high-quality 4D-STEM data in the revised manuscript further strengthens its technical merit, and in my view, the work meets the standards of Nature Communications in this respect.”*

Response: The main criticism of Reviewer #2 regarding the novelty of our work still seems to stem from a misunderstanding of the 3D ED approach we employed, as already apparent from the first review round. Upon careful examination of the literature cited by Reviewer #2, we found that the referenced studies differ entirely in their objectives, scope, and experimental approach. In these works, the authors examine thin epitaxial films with large crystalline domains—fabricated by directional epitaxial crystallization—and tilt them in discrete zone axes to determine their lattice parameters and epitaxial crystallization direction. This is not even 3D ED, but conceptually in line with the well-established field of 3D ED that deals with crystal structure analysis of single nanocrystals (Gemmi et. al., ACS Central Science 5 (2019) 1315, ref. [36] in our manuscript). Our approach is fundamentally different: we use 3D ED, for the first time, to study the texture and texture evolution of nanocrystalline organic films containing thousands of nanocrystals in the diffraction volume, characteristic for bulk heterojunction (BHJ) solar cells. We demonstrated the generality of 3D ED by analysing two different OSC systems and highlighted the integration of our 3D ED approach with other TEM techniques. To our knowledge, no prior work has achieved this level of crystallographic analysis in BHJ films. Thus, the manuscript’s novelty and the broader applicability of 3D ED in the OSC field remain clear and significant. This perspective is also clearly supported by Reviewer #1, reinforcing the innovation presented in our work. We hope this explanation addresses and resolves Reviewer #2’s concerns regarding the unique aspects of our approach.

2.Comment Reviewer #2: *“While I acknowledge that GIWAXS has directional limitations and cannot provide full crystallographic information, 3D ED similarly fails to deliver 3D real-space nanomorphology in OSC studies. For example, Figure 4 only illustrates a structural transition from random orientation (diffraction rings) to anisotropic orientation (diffraction arcs) via tilt angle adjustment, without clarifying how the small molecules pack in 3D or in thin films. No illustrative model of molecular arrangement is provided neither. Moreover, confirming edge-on or face-on configurations via 3D ED, which is as also effectively accomplished by GIWAXS in Figure 5, does not offer substantial additional insight.”*

Remark Reviewer #1: *“For comment 2, the additional 4D-STEM data partially addresses Reviewer #2’s concerns, as nanoscale orientations can be distinguished by 4D-SCED, which is not achievable by GIWAXS alone. This highlights the value of correlative STEM analysis that integrates 3D ED with other STEM methods. However, I concur with Reviewer #2 that the manuscript would benefit from illustrative models of molecular arrangements to enhance accessibility for a broader readership.”*

Response: We wish to clarify that our 3D ED approach is not intended to “deliver 3D real-space nanomorphology”, as suggested by Reviewer #2, while we do not oppose that this is crucial for device performance, as 3D real-space morphology and phase separation determine charge carrier transport paths to interfaces and electrodes. As stated in the original introduction, 3D ED integrated

in a correlative TEM workflow provides “3D molecular ordering parameters, such as texture and mosaicity” (page 4, lines 103-104), enriching the characterization of nanomorphology, in-plane molecule ordering, local molecule packing, and composition with additional **structural** parameters. In contrast to GIWAXS, 3D ED can bridge the gap to real space by leveraging the imaging and analytical capabilities of a TEM – an aspect recognized and appreciated by Reviewer #1. The integration of 3D ED in a comprehensive TEM workflow allows for a direct correlation of real- and reciprocal space information within the same sample. In the last revision of our manuscript, we included extensive correlative analyses (STEM-EELS composition mapping and 4D-SCED analysis, cf. Figure 6 of revision 1, now Figure 5) to directly link local molecular packing with the nanoscale morphology – an achievement not possible with GIWAXS alone. 3D ED not only provides diffraction information equivalent in content to GIWAXS but does so (i) on a much more local scale and (ii) in direct correlation with real-space information—all within a single instrument, a modern analytical TEM available in many laboratories worldwide. If this is indeed considered not to be of interest to the OSC community, it is difficult to see what would be.

We thank Reviewer #2 and #1 for the second point raised here, pointing out the need for more illustrative models of molecular arrangement. As the molecular arrangement of DRCN5T and P3HT in crystallites has been extensively discussed in research articles aiming for structure solution,^{1,2} we based our discussion of results on those available structures. However, we agree that including additional schemes would facilitate better understanding in the broad readership of Nature Communications, where not all readers may be aware of typical crystallographic structures in organic semiconductors and how exactly they are represented in different diffraction methods.

Changes in manuscript: Based on these remarks, we made the following adjustments both in Figures and describing text, convinced that they further strengthen our manuscript:

- MS Figure 2 (new Figure 2, based on Figure 2 and 3 of last revision, see Comment #3 below): We adjusted the inset of the exemplary crystallite and the figure caption to clarify typical molecular arrangements in organic semiconductor crystallites in order to facilitate clarity for later on following crystal alignment schemes. In the main text we explicitly mention ‘*For organic semiconductors ordering of the molecules in crystallites is typically described with respect to three directions, namely π - π stacking as [010], lamellar stacking as [100] and backbone direction as [001],¹⁹ which is also applied in the following (cf. inset in Figure 2a).*’ (page 6, lines 132-134)
- MS Figure 3 (new Figure 3, based on original Figure 4, 3D ED data and processing): We now explicitly show the arrangement of DRCN5T in crystallites and the typical preferred crystallite orientations edge-on and face-on. Furthermore, we specify with numbered color-code markings how inclinations and rotations of those orientations are represented in the 3D ED data. In the figure caption we describe as follows (page 10-11, lines 234-244): ‘**Figure 1. Structural character of model molecular crystal and corresponding reciprocal space features extracted from 3D ED. a** The molecular arrangement of DRCN5T is exemplary shown for an edge-on crystallite. Different crystallite orientations and inclinations with respect to the electron beam are represented in the 3D ED data as marked accordingly: in-plane rotation of i) edge-on and ii) face-on crystallites, and iii) inclination.

b A tilt series from -78° to 80° was acquired in 1° steps. The pattern recorded at 0° sample tilt angle (in-plane) indicates the presence of both edge-on and face-on crystallites with random in-plane orientation (i and ii). *c* The representative patterns from the tilt series at 30° and 75° sample tilt reveal the texture of the system, manifested intense diffraction ring segments. The bright segments of the outermost π - π stacking (010) ring appearing around the q_y -axis represent edge-on crystallites, whereas the otherwise weaker, continuous (010) ring indicates a fraction of isotropic crystallites (iii). *d* Out of the tilt series the three-dimensional reciprocal space volume is reconstructed, incorporating diffraction information of the differently oriented crystallites. The ortho-slice q_y - q_z plane is offset from $q_x = 0$ to avoid the missing wedge. For comparison to GIWAXS an azimuthal integration is performed (see Figure 4a).’ The schemes are mentioned in the main text accordingly (page 9, lines 206-207 and 215-220).

- MS Figure 5 (adapted Figure 6 of last revision, DRCN5T TEM correlative results): clarifying color wheel representation of 4D-SCED data with respect to edge-on crystallite in-plane rotation, addition of schematic illustration of arrangement of crystallites in domains based on correlative TEM workflow and illustration of crystallite information included in 3D ED. The figure caption describes as follows: ‘A schematic arrangement of edge-on crystallites within a leaf-like domain based on the correlative TEM analysis is depicted as inset, highlighting the mosaic structure and π - π stacking direction along the short axis of the domain. [...] c [...] 3D character of diffraction information available from 3D ED, including crystallite in-plane-rotation (Φ), out-of-plane-inclination (χ) and CCL. ...’ (page 15, lines 341-345)
- MS Figure 6 (enriched Figure 7 of last revision, P3HT results): Incorporation of a dedicated Figure panel for schematic representation of molecular arrangement of P3HT for edge-on and face-on regions based on correlative TEM analysis, described as follows in the caption: ‘Scheme of molecular structure of P3HT and edge-on and face-on ordered regions. While the structure information in the substrate plane can be directly observed from 4D-SCED analysis, the out-of-plane information can be deduced from CCLs analysis of 3D ED.’ (page 17, lines 394-496) (for more details concerning additional measurements, results and discussion cf. response to comment #4)

3.Comment Reviewer #2: “The writing and organization of the manuscript remain exceedingly poor. At times, it reads more like a review article than a research paper. The discussion relies heavily on common knowledge, which are readily accessible via Google or general literature, rather than on critical analysis of the experimental results. Furthermore, the manuscript is unnecessarily lengthy, yet contains limited substantive data, or original insight. Both in content and structure, it falls short of the standards expected of a high-quality research publication.”

Remark Reviewer #1: “With respect to comment 3, I fully agree with Reviewer #2. While this does not affect the technical robustness of the work, clear writing and organization are crucial for communicating the findings to non-specialist audiences. The manuscript currently reads more like

a perspective or methodology article than a focused research paper. Additionally, the figures in the main text are not optimally composed, consolidating the eight figures into four or five with improved clarity would be beneficial.”

Response: We regret that Reviewer #2 had a negative impression of writing and structure, and are additionally somewhat surprised by the remark of Reviewer #1. In the first round neither Reviewer #1 nor #3 indicated, that the paper was too long or unfocused. In fact, Reviewer #1 explicitly praised the high-quality new data and fully addressed concerns, noting that the level of detail was “...illuminating for seasoned experts and newcomers alike”. Reviewer #3 also highlighted the “... level of detail for others to adopt this approach” and states that “the manuscript is well written and clearly presented”. In the first round of revision, we significantly restructured the manuscript, including additional data and discussion of results, particularly regarding the correlative data of DRCN5T and 3D ED of two P3HT samples. Additionally, we want to emphasize that the expansion of the discussion section was driven by Reviewer queries, such as the method’s applicability to other material systems and its time and spatial resolution. We gladly provided more insights into those topics to meet the apparent need for detailed explanations on broader applicability (page 20-22, lines 479-532), intending to help general readers grasp the context and importance of 3D ED. We are confident that we present our approach in a convincing and structured manner, driven by experimental results, while providing necessary background and discussing broader applicability – all aligned with Nature Communications’ broad readership.

We believe the perception of our paper as lacking insight may stem from Reviewer #2’s initial mis-judgment of the novelty and hope our response to comment #1, combined with the supporting remark from Reviewer #1 to comment #1, resolves this issue.

Not least due to the first round of revision and also the apparent misunderstandings of Reviewer #2, we do not find it reasonable to shorten the manuscript excessively in background and discussion section, as Reviewer #2 seems to suggest. Nonetheless, where we identified opportunities to enhance clarity, with the aim to communicate our findings as clearly and concisely as possible, we have refined text, condensed figures and (also with respect to comment #4 below) included additional measurements based on the Reviewer comments.

Changes in manuscript:

- MS Figure 2 (new Figure 2, based on Figure 2 and 3 of last revision): The figure consolidates Figure 2 and 3 of the last revision, enabling a closer integration of scattering geometry and corresponding reciprocal space sampling of 3D ED and GIWAXS. The figure caption and description in main text were adjusted accordingly, see for example Figure caption on pages 7-8, lines 154-167.
- Main text: In order to facilitate a more concise description of real and reciprocal space setup presented in new Figure 2, minor redundancies were removed and text passages rearranged. This includes for example limiting discussion on footprint and inelastic scattering contributions to discussion section (page 18, lines 412-424) and removing redundancy on the large Ewald sphere using electrons. Furthermore, we adjusted wording

in lines 135-151 to move focus more specifically on 3D ED compared to conventional plan-view ED. Additional adjustments are marked in the manuscript accordingly (e.g. page 8, lines 169-190).

- MS Figure 6 (new Figure 6, based on Figure 7 of last revision) and SI Figures S8, S9, S10 and Tables S2 and S3: We included additional data on the P3HT system, integrating our 3D ED findings more closely in a correlative TEM workflow, as detailed in our response to Comment #4 below. In this context we expanded discussion of our results and included schemes of molecular arrangements based on our findings (Comment #2) (page 16-17, lines 359-396).

4.Comment Reviewer #2: *“The manuscript merely posits that 3D ED can be used as a tool for studying OSCs, without justifying why this method is necessary or what novel insights it can provide. Key questions remain unaddressed: What unprecedented discoveries does it enable in OSC research? How can it reveal new mechanisms, enhance material performance, or provide design principle of OSC materials? Without discussing these innovative aspects, the paper is unlikely to attract broad interest from the readership.”*

Remark Reviewer #1: *“Addressing comment 4, I appreciate Reviewer #2’s concern regarding the necessity and novelty of the 3D ED method. Although the authors have included a comparison between pristine and annealed specimens, this alone does not sufficiently demonstrate a comprehensive mechanism of structural transition. I strongly recommend that the authors perform in situ annealing within the TEM, capturing both 3D ED and real-space imaging, as well as 4D-SCED datasets, to validate both structural and nanoscale morphological changes. At a minimum, ex situ correlative analyses using 3D ED, STEM imaging, and 4D-SCED on specimens before and after annealing should be conducted, rather than relying solely on 3D ED.”*

Response:

We respectfully, yet firmly disagree with Reviewer #2. In our manuscript we showcase why 3D ED is needed and valuable using experimental data. The integration of our 3D ED approach in a comprehensive TEM workflow enabled correlation of multimodal data in the DRCN5T system: while 3D ED allowed for resolving CCL, texture and mosaicity, STEM-EELS revealed phase separation and 4D-SCED a correlation between local molecular packing and (projected) domain shape. Only this close interplay of information enabled a clear picture of preferred crystallite orientations in a mosaic domain structure with a correlation of domain shape and molecular packing. We explicitly point out effects on charge transport pathways and relevance for understanding structure-property relationships. (page 14, lines 314-319 and page 15, lines 333-335 in revised manuscript) Furthermore, by using 3D ED, we revealed the change of 3D crystallographic orientation of the P3HT active layer upon thermal treatment (randomly oriented nanocrystals becoming preferentially oriented). Such findings illustrate how 3D ED can uncover nanoscale structure that directly relate to OSC performance (e.g., how domain orientation can affect charge transport pathways).

While we showcased the broader applicability of our 3D ED approach to clearly resolve structural change in P3HT upon thermal annealing, we respect the query of Reviewer #1 to once again integrate in the comprehensive TEM workflow for correlation with nanomorphology and local packing from the *ex situ* annealed samples. We hope these additions will also convince Reviewer #2. We note that *in situ* correlative observations is solely limited by the current availability of TEM holders that allow simultaneous heating, large observation area and high tilting range, which does not influence the principle of correlative TEM analysis discussed in the manuscript.

We agree that articulating broader implications is important, and we now highlight more explicitly how insights from 3D ED in a comprehensive TEM workflow could inform material design and processing for improved OSC efficiency. However, we remain convinced that the data and discussions already present in the manuscript clearly demonstrate novel insights enabled by 3D ED and correlative TEM characterization, not merely the application of a known tool.

Changes in manuscript:

- MS Figure 6 (new Figure 6, based on Figure 7 of last revision): The figure previously composed of 3D ED data now integrates additional 4D-SCED measurements and schematic representations of P3HT crystallite alignment.
- SI Figure S8 (expanded Figure S8 of last revision): now also includes out-of-plane intensity profiles, to consolidate the comprehensive information included in 3D ED
- SI Table 2 (new): summarizes CCLs based on in-plane and out-of-plane profiles of 3D ED to emphasize 3D ED provides quantitative results on 3D molecular ordering
- SI Figure S9: summarizes beam damage behaviour of P3HT:PC₇₁BM, as detailed in the figure caption: '**Figure S9. Beam damage behavior of P3HT:PC₇₁BM system WO and TA 10min under the electron beam. a,b** A series of diffraction patterns was acquired at one position, capturing the degradation behavior. The diffraction patterns were polar transformed and projected along azimuth axis, resulting in a *q-t* (i.e., *q-dose*) map depicting diffraction intensity as function of accumulated dose. **c,d** Linecuts of 100 and π - π stacking peak were extracted along accumulated dose, showing the decreasing intensity of diffraction peaks.'
- SI Table S3: summarizing evaluated critical doses for P3HT:PC₇₁BM system
- SI Figure S10: correlative STEM and EFTEM measurements of P3HT:PC₇₁BM system, showing the coarsening of nanomorphology and enhanced phase separation upon thermal annealing.
- Main text: all changes indicated in the list above are integrated into the MS, incorporating all correlative results, also from SI, closely into the main text. (pages 16-17, lines 351-396)
- Main text: broader implications are more explicitly highlighted, e.g. in the introduction (page 5, lines 105-107), and results (page 15, lines 328-329 and 334).

Final Remarks

We want to thank the reviewers and editor for their time and consideration. We hope our revisions and this response are convincing and resolve previous reservations.

References

- 1 Berlinghof, M. *et al.* Crystal-structure of active layers of small molecule organic photovoltaics before and after solvent vapor annealing. *Zeitschrift für Kristallographie - Crystalline Materials* **235**, 15–28 (2020). <https://doi.org/10.1515/zkri-2019-0055>
- 2 Kayunkid, N., Uttiya, S. & Brinkmann, M. Structural Model of Regioregular Poly(3-hexylthiophene) Obtained by Electron Diffraction Analysis. *Macromolecules* **43**, 4961-4967 (2010). <https://doi.org/10.1021/ma100551m>

We thank the Reviewers for their careful evaluation of our manuscript and for their constructive and insightful comments. We are pleased that we have been able to address all points raised and believe that the revisions have improved the manuscript.